# When shared concept cells support associations: Theory of overlapping memory engrams

**Chiara Gastaldi**[1]*, **Tilo Schwalger**[2‡], **Emanuela De Falco**[3], **Rodrigo Quian Quiroga**[4,5], **Wulfram Gerstner**[1‡]

**1** School of Computer and Communication Sciences and School of Life Sciences, École Polytechnique Fédérale de Lausanne, Lausanne, Switzerland, **2** Institut für Mathematik, Technische Universität Berlin, Berlin, Germany, **3** School of Life Sciences, École Polytechnique Fédérale de Lausanne, Lausanne, Switzerland, **4** Centre for Systems Neuroscience, University of Leicester, Leicester, United Kingdom, **5** Peng Cheng Laboratory, Shenzhen, China

‡ These authors jointly supervised this work.
* chiara.gastaldi@epfl.ch

**Data Availability Statement:** All relevant data are within the manuscript and its Supporting information files. The code used to numerically solve the equations derived in the manuscript is

## Abstract

Assemblies of neurons, called concepts cells, encode acquired concepts in human Medial Temporal Lobe. Those concept cells that are shared between two assemblies have been hypothesized to encode associations between concepts. Here we test this hypothesis in a computational model of attractor neural networks. We find that for concepts encoded in sparse neural assemblies there is a minimal fraction $c_{min}$ of neurons shared between assemblies below which associations cannot be reliably implemented; and a maximal fraction $c_{max}$ of shared neurons above which single concepts can no longer be retrieved. In the presence of a periodically modulated background signal, such as hippocampal oscillations, recall takes the form of association chains reminiscent of those postulated by theories of free recall of words. Predictions of an iterative overlap-generating model match experimental data on the number of concepts to which a neuron responds.

## Author summary

Experimental evidence suggests that associations between concepts are encoded in the hippocampus by cells shared between neuronal assemblies ("overlap" of concepts). What is the necessary overlap that ensures a reliable encoding of associations? Under which conditions can associations induce a simultaneous or a chain-like activation of concepts? Our theoretical model shows that the ideal overlap presents a tradeoff: the overlap should be larger than a minimum value in order to reliably encode associations, but lower than a maximum value to prevent loss of individual memories. Our theory explains experimental data from human Medial Temporal Lobe and provides a mechanism for chain-like recall in presence of inhibition, while still allowing for simultaneous recall if inhibition is weak.

available at https://github.com/ChiaraGastaldi/pub_Gastaldi_2021_AttractorNetwork.git.

**Funding:** WG and CG were supported by the Swiss National Science Foundation (www.nsf.gov), grant agreement 200020_184615 and by the European Union Horizon 2020 Framework Program (https://ec.europa.eu/programmes/horizon2020/) under agreement no. 785907 (HumanBrain Project, SGA2). RQQ acknowledges support from Biotechnology and Biological Sciences Research Council. The funders had no role in study design, data collection and analysis, decision to publish, or preparation of the manuscript.

**Competing interests:** The authors have declared that no competing interests exist.

## Introduction

Human memory exploits associations between concepts. If you visited a famous place with a friend, a postcard of that place will remind you of him or her. The episode "with my friend at this place" has given rise to an association between two existing concepts: before the trip (the episodic event), you already knew your friend (first concept) and had seen the place (second concept), but only after the trip, you associate these two concepts.

Concepts are encoded in the human Medium Temporal Lobe (MTL) by neurons, called "concept cells", that respond selectively and invariantly to stimuli representing a specific person or a specific place [1–3]. Each concept is thought to be represented by an assembly of concept cells that increases their firing rates simultaneously upon presentation of an appropriate stimulus. The fraction $\gamma$ of neurons in the human MTL which is involved in the representation of each concept is estimated to be $\gamma \sim 0.23\%$ [4]. Under the assumption that each memory item is represented by the activation of a fixed, but random, subset of active neurons, a single concept is expected to activate $\gamma N$ neurons and two arbitrary concepts are expected to share $\gamma^2 N$ cells, where $N$ is the total number of neurons in the relevant brain areas.

Experimental studies have shown that single neurons can become responsive to new concepts while learning pairs of associations [5]. Moreover, it has been estimated that assemblies representing two arbitrary concepts share less than 1% of neurons, whereas assemblies representing previously associated concepts share about 4–5% of neurons [6] suggesting that an increased fraction of shared neurons supports the association between concepts [6–8].

With the presence of shared neurons, the activation of a first assembly (e.g., a place) may also activate a second assembly (e.g., a person). This poses several theoretical questions. First, for the brain to function correctly as a memory network, it must remain possible to recall the two associated concepts separately (e.g. place without your friend), and not automatically the two together. However, if the concepts share too many neurons it becomes likely that the two memory items can no longer be distinguished, but are merged into a single, broader concept encoded by a larger number of active neurons. We therefore ask as a first question: what is the maximally allowed fraction $c_{max}$ of shared neurons between two assemblies before the possibility of separate memory recalls breaks down? Shared concept cells can be visualised as an overlap between two memory engrams. Below the maximal fraction $c_{max}$ of shared neurons, each of the associated patterns can be recalled as a separate memory pattern, as schematically illustrated in Fig 1A.

As an alternative to a *static* recall of one or the other concept (or the two associated concepts together), we could also ask whether the activation of a concept would facilitate the recall of an associated one, or even a *temporal* chain activation of associations (as described in free memory recall tasks [9–12]), due to overlaps in the representations. In this context, we ask a second question: if each concept is represented by a small fraction of active neurons $\gamma$, given the activation of a concept, is there a minimal fraction of shared neurons $c_{min}$ necessary to enable a reliable activation of associated ones?

Moreover, while most experimental studies have dealt with pairwise associations between, say one person and one place, more recent work has shown that a single neuron can respond to multiple concepts [6], e.g., several related places. In view of this, we ask a third question: how should memory be organized in a neural network such that $k$ different memory engrams all have the equal size pairwise overlaps?

Associative memory in recurrent networks, such as the area CA3 of the hippocampus, has been modeled with attractor neural networks [13–17] where each memory item is encoded as a memory engram [18, 19] in a fixed random subset of neurons (called "pattern" in the theoretical literature [17]) such that no pattern has an overlap above chance with another one.

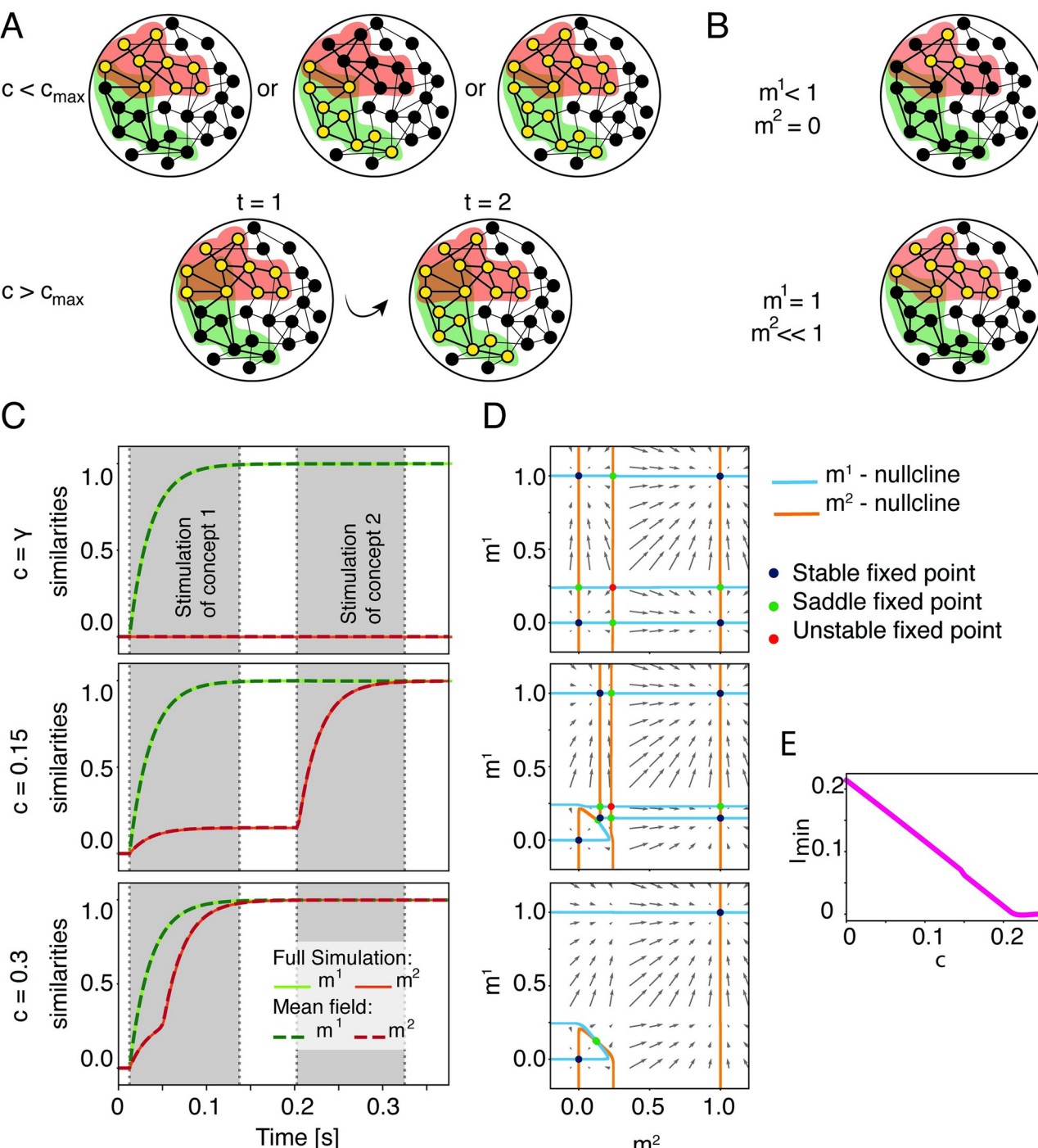

**Fig 1. Overlapping concepts can be retrieved separately and jointly.** A) Activation of concepts (schematic). Black filled circles = inactive neurons. Yellow filled circles = active neurons. Colored halos (red, green) represents assignment to a specific concept. When the fraction of shared neurons is small (top row, $c < c_{max}$) the two concepts can be recalled separately or together. If the number of shared concept cells is too large (bottom row, $c > c_{max}$), the recall of a first concept (red) leads inevitably to the activation of the second associated concept (green). B) Similarity measure. If only a subset of neurons belonging to the first memory engram is activated (top), the configuration exhibits similarities $m^1 < 1$ and $m^2 = 0$. If the first memory is fully recalled, while memory 2 is not (bottom), the similarity measures are $m^1 = 1$ and $m^2 \ll 1$. C) Dynamics of the similarities for different fractions of shared neurons. The similarities $m^1$ (green) and $m^2$ (red) as a function of time in a full network simulation (solid lines) are compared to predictions of mean-field theory (dashed lines). Strong external stimulation $I_1 = 0.3$ is given to the units belonging to concept $\mu = 1$ during a first stimulation period and a weak external stimulation $I_2 = 0.1$ is given to the units belonging to concept $\mu = 2$ during the second stimulation period (in grey). If $c > c_{max}$, the concept 2 gets activated without receiving any stimulation. D) Three phase-planes of the dynamics of similarity variables $m^1$ and $m^2$ for different values

of fraction of shared neurons $c$. Arrows indicate direction and speed of increase or decrease of the similarity variables. Intersections of blue and orange lines (the "nullclines" of the two variables $m^1$, $m^2$) indicate fixed points, with a stability encoded by color (legend). E) Minimum amplitude of the external stimulation $I^2$ needed to activate the memory of the second concept if the first one is activated (as a function of the fraction of shared neurons $c$). Parameters: $\hat{h}_0 = 0.25$, $\hat{b} = 100$, $r_{\max} = 40$ Hz, $\tau = 25$ ms, $\alpha = 0$, $\gamma = 0.2\%$. For simulations: $N = 10000$, $P = 2$.

Animal studies provide evidence of attractor dynamics in area CA3 [20, 21]. The few theoretical studies that considered overlapping memory engrams above chance level in the past [22, 23] focused on overlaps arising from a hierarchical organization of memories. Whereas such a hierarchical approach is suitable for modeling memory representation in the cortex, we are interested in modeling MTL, and in particular area CA3 of the hippocampus, where experimentally no hierarchical or topographical organization has been observed [6]. In experiments, episodic associations between *arbitrary* different concepts (such as a person and a place)—and shared neurons in the corresponding assemblies—can be induced by joint presentation of images representing the different concepts [5]. Inspired by these experiments, we create pairwise associations between a number of concepts by artificially introducing shared concept cells in the model. We will talk about "overlapping engrams" if the number of shared concept cells is beyond the number $\gamma^2 N$ of cells that are shared by chance.

## Results

The first two questions introduced above can be summarized as a more general one: What is the role of those concept cells that are shared between stored memory engrams? To answer this question, we consider an attractor neural network of $N$ neurons in which $P$ engrams are stored in the form of binary random patterns [7]. The pattern $\vec{\xi}^\mu = \{\xi_i^\mu \in \{0, 1\}; 1 \leq i \leq N\}$ with pattern index $\mu \in \{1, \ldots, P\}$ represents one of the stored memory engrams: a value $\xi_i^\mu = 1$ indicates that neuron $i$ is part of the stored memory engram and therefore belongs to the assembly of concept $\mu$, while a value of $\xi_i^\mu = 0$ indicates that it does not. A network that has stored $P$ memory engrams is said to have a memory load of $\alpha = P/N$.

Since concept-cells in human hippocampus form sparse neural assemblies with a sparseness parameter $\gamma \sim 0.23\%$ [4], we focus on the case of sparse memory engrams. In other words, an arbitrary neuron $i$ has a low probability $\gamma = \text{Prob}(\xi_i^\mu = 1) \ll 1$ to participate in the assembly of concept cells corresponding to memory engram $\mu$.

The attractor neural network is implemented in a standard way [24, 25]. Each neuron, $i = 1, \cdots, N$, is modelled by a firing rate model [25]

$$\tau \frac{dr_i}{dt} = -r_i + \phi(h_i), \tag{1}$$

where $r_i(t)$ is the firing rate of neuron $i$ and $\phi(h) = r_{\max}/\{1 + \exp[-b(h - h_0)]\}$ is the sigmoidal transfer function, or frequency-current (f-I) curve, characterized by the firing threshold $h_0$, the maximal steepness $b$, and the maximal firing rate $r_{\max}$. The patterns $\vec{\xi}^\mu$ are encoded in the synaptic weights $w_{ij}$ via a Hopfield-Tsodyks connectivity for sparse patterns so that the average of synaptic weights across a large population of neurons vanishes [17].

In attractor neural network models, memory engrams $\mu$ induce stable values $r_{\mu,i}^*$ of the neuronal firing rates during the retrieval of a stored concept. In mathematical terms, to each engram $\mu$ corresponds a fixed point $\vec{r}_\mu^*$ in such a way that the firing rate $r_{\mu,i}^*$ of neuron $i$ is high if $\xi_i^\mu = 1$ and low if $\xi_i^\mu = 0$. When the network state $\vec{r}(t)$ is initialized close enough to the stored memory $\mu$, the attractor dynamics drives the network to the retrieval state $\vec{r}_\mu^*$ characterized by persistent activity of all those neurons that belong to the assembly of concept $\mu$.

The similarity between the momentary network state and a stored memory $\mu$ is defined as

$$m^{\mu}(t) = \frac{1}{N\gamma(1-\gamma)r_{\max}} \sum_{j=1}^{N} \left( \xi_j^{\mu} - \gamma \right) r_j(t). \qquad (2)$$

The similarity measures the correlation between the firing rates $\{r_j(t)\}_{j=1,\ldots,N}$ and the stored patterns $\vec{\xi}^{\mu}$ such that if memory concept $\mu$ is retrieved, then $m^{\mu} \sim 1$ (schematics in Fig 1B), and, if no memory is recalled (*resting state*), then $m^{\mu} \sim 0$ for all $\mu$. The similarity of the network activity with a stored memory develops as a function of time. For example, computer simulations of a network of $N = 10000$ interacting neurons indicate that, if one of two engrams that share concept cells is stimulated for 120ms, then the similarity of the network activity with this engram increases to a value close to one, indicating that the memory has been recalled (Fig 1C middle) while the second memory is only weakly activated quantified by a small, but non-zero similarity. However, if the fraction of shared neurons is above a maximally allowed fraction $c_{\max}$, then the second memory always gets activated even before it is stimulated (Fig 1C bottom) indicating that associations are so strong that the two concepts have been merged.

## Maximal fraction of shared neurons between memory engrams

In order to better understand the network dynamics, we develop a mathematical theory that depends on the fraction of neurons $c$ that are shared between two engrams. The *total number n of shared neurons* in a network of size $N$ depends on $c$ and the sparsity parameter $\gamma$ via the relation $n = \gamma c N$.

Let us imagine to gradually increase the fraction of shared neurons between the first two memory engrams. At the lowest end, $c = \gamma$, the patterns $\vec{\xi}^1$ and $\vec{\xi}^2$ are independent, and hence cell assemblies 1 and 2 share a small fraction of neurons corresponding to chance level. It is well known, that in this case, each memory engram generates a separate attractive fixed-point of the network dynamics [17] implying that the two corresponding concepts can be retrieved separately. However, experimental data reports that, for associated concepts, the fraction of shared neurons $c \sim 4$–5% [6] is much larger than the chance level $\gamma \sim 0.23\%$. This observation suggests that the patterns $\vec{\xi}^1$ and $\vec{\xi}^2$ of two associated memory engrams have a fraction of shared neurons larger than chance level, $c > \gamma$. On the other hand, in the (trivial) limit case of large fraction of shared neurons $c \to 1$, the two memory engrams and hence the two cell assemblies share all neurons, and it is clearly impossible to retrieve one memory without the other.

To study the maximal fraction of shared neurons $c_{\max}$ at which independent memory recall breaks down, we use a mean-field approach for large networks and work in the limit $N \to \infty$. In this limit, it is possible to fully describe the network dynamics using the similarities $m^{\mu}$ as the relevant macroscopic variables. Since we are interested in the retrieval process of concepts $\mu = 1$ and 2, we assume the similarity of the present network state with other memories $\mu > 2$ to be close to zero: we will refer to these non-activated memories as "background patterns". Under these assumptions, we find dynamical mean-field equations that capture the network dynamics through the similarity variables $m^1$ and $m^2$.

$$\tau \frac{dm^1}{dt} = -m^1 + F_1(m^1, m^2) \qquad (3a)$$

$$\tau \frac{dm^2}{dt} = -m^2 + F_2(m^1, m^2) \qquad (3b)$$

where the explicit form of the functions $F_1$ and $F_2$ is given in Eq (44) of Materials and methods (or Eq (45) for the special case of small load $\alpha$). Eq (3) represents a two-dimensional dynamical systems which can be analyzed using phase-plane analysis. Fig 1D shows three phase-planes in the $m^1 - m^2$ space, each for a different value of the fraction of shared neurons. The $m^1$- or $m^2$-nullclines solve $dm^1/dt = 0$ or $dm^2/dt = 0$ in Eqs (3a) and (3b), respectively. The intersections between the $m^1$- and $m^2$-nullcline are equilibrium solutions, or fixed points, of the mean-field dynamics and are color-coded according to their stability. For $c = \gamma$, we identify four stable fixed points: the resting state $(m^1, m^2) = (0, 0)$, two single-retrieval states $(m^1, m^2) = (1, 0)$ and $(m^1, m^2) = (0, 1)$ corresponding to the retrieval of concept $\mu = 1$ and the retrieval of concept $\mu = 2$, respectively. Finally, there is a symmetric state which corresponds to the activation of both concepts simultaneously, $(m^1 = m^2 \lesssim 1)$.

Once a maximally allowed value $c = c_{\max}$ is reached, the two single-retrieval states merge with their nearby saddle points and disappear. To compute the numerical value of the maximal fraction of shared neurons, we extract it following the procedure described in the paragraph "Extract numerically the maximal correlation" in the MATERIALS AND METHODS. For fractions of shared neurons $c > c_{\max}$ only two stable fixed points are left, the resting state and the symmetric state in which assemblies of both concepts are activated together: this symmetric state is the theoretical description of the state that we qualitatively predicted above where the activation of a first concept leads inevitably to the activation of the second, overlapping one (Fig 1C, bottom). The minimum external stimulation needed to activate the second concept depends on the fraction of shared neurons (Fig 1E). With our choice of parameters, no external stimulation is needed to recall the second memory, if the fraction of shared neurons is $c > c_{\max} = 22\%$, since the two concepts have merged into a single one and are *always* recalled together.

In the limit of infinite steepness $b \to \infty$, vanishing load $\alpha = 0$ and vanishing sparseness $\gamma \to 0$, the value $c_{\max}^0$ of the maximal fraction of shared neurons can be calculated analytically. Since this value provides an upper bound of the maximal fraction of shared neurons for arbitrary $b$, we have the inequality (Fig 2A)

$$c_{\max} \leq c_{\max}^0 \equiv \gamma + (1 - \gamma)\frac{h_0}{A\,r_{\max}}, \tag{4}$$

where $A$ characterizes the overall strength of synaptic weights (see Eq (5) below). Further analysis (see Materials and methods) shows that the stationary states of the mean-field dynamics depend—apart from the parameters $\gamma$, $C$ and $\alpha$ related to the patterns—only on two dimensionless parameters: the rescaled firing threshold $\hat{h}_0 = h_0/(A\,r_{\max})$ and the rescaled steepness $\hat{b} = b \cdot (A\,r_{\max})$. We find that the maximal fraction of shared neurons $c_{\max}$ increases with $\hat{b}$ and also with $\hat{h}_0$ (Fig 2A).

We proceed by studying how the maximal fraction of shared neurons varies as a function of the memory load $\alpha = P/N$ (Fig 2B). As the load increases, we observe that the maximal fraction of shared neurons decreases, but the change is modest. This weak dependence on the load is robust against two variations of the network where (i) self-interaction of neurons is excluded; or (ii) the $P - 2$ background patterns are also overlapping in pairs, e.g., pattern 3 is overlapping with pattern 4, 5 with 6, etc. For both modifications, the mean-field equations look slightly different (Materials and methods) but neither modification leads to a significant change of the maximal fraction of shared neurons $c_{\max}$ (Fig 2A). In a network that has stored a total of $P$ memory engrams, the maximal fraction of shared neurons could potentially depend on the group size $p$ of patterns that are all overlapping with each other. So far we have considered $p = 2$. We extended the mean-field approach to the case of three and four overlapping patterns (SI) by rewriting and adapting Eq (44) in MATERIALS AND METHODS. Again we find that the

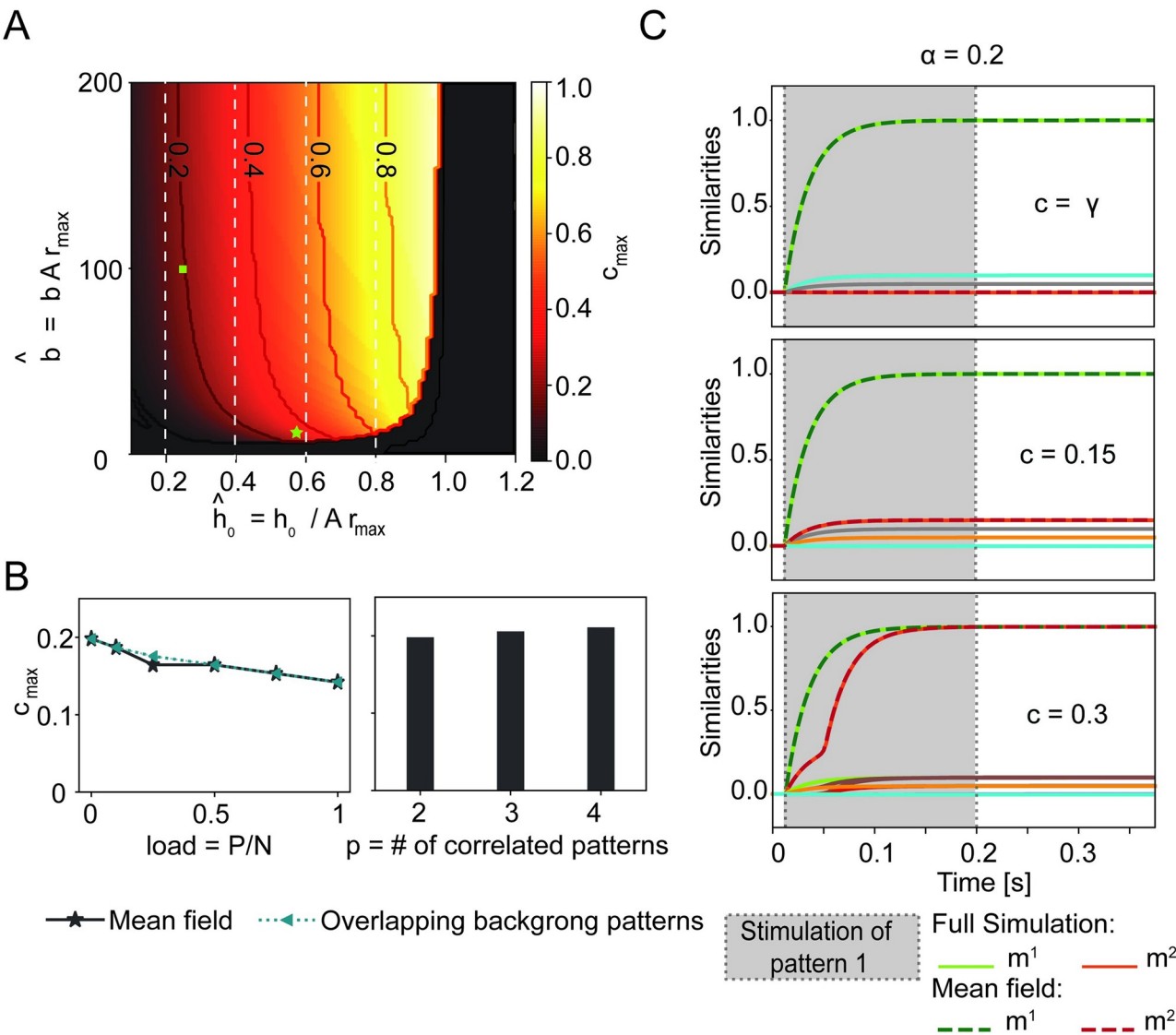

**Fig 2. The maximal fraction $c_{max}$ of shared neurons depends on the neuronal frequency-current curve but not on the memory load.** A) Maximal fraction ($c_{max}$, color code) as a function of the parameters $\hat{b} = b\,Ar_{max}$ (steepness) and $\hat{h}_0 = h_0/(Ar_{max})$ (firing threshold) of the frequency-current curve. Niveau lines added for indicated values of $c_{max}$. In the black area the resting state is the only stable solution. Vertical white dashed lines indicate the theoretical upper bound $c_{max}^0$, for different values of $\hat{h}_0$. The green square indicates the parameter choice used in Figs 1 and 2B and 2C. The green star indicates the parameters extracted for the Macaque inferotemporal cortex [24]. B) Maximal fraction $c_{max}$ of shared neurons as a function of the memory load $\alpha = P/N$ (left graph) without (solid grey line) or with overlaps in pairs of two of the $P - 2$ background patters (dashed green line); and as a function of the number $p$ of correlated patterns (histogram, right graph). C) As in Fig 1C, but with a large number of background patterns ($\alpha = 0.2$). Network activity exhibits only small similarity with background patterns (diversely colored lines) but large similarity with the stimulated pattern $\mu = 1$. Parameters (unless specified): $\gamma = 0.2\%$, $\hat{b} = 100$, $\hat{h}_0 = 0.25$, $r_{max} = 40$ Hz, $\tau = 25$ms; $\alpha = 0$ in A-B. For simulations in C: $N = 10000$, $p = 2$, $\gamma = 0.2\%$.

maximal overlap is not significantly influenced by the group size $p$ of overlapping patterns (Fig 2B). The group size $p$ can be large provided that the total number of patterns $P$ does not exceed the memory capacity of the network.

In summary, we found a maximal fraction $c_{max}$ of shared neurons beyond which the retrieval of single concepts is no longer possible. The value of $c_{max}$ depends on frequency-current curve of neurons.

## What is the minimal fraction of shared concept cells to encode associations?

We find that a symmetric double-retrieval state exists where two concepts are recalled at the same time (Fig 1D, top), even if the fraction of shared concept cells is at chance level. This co-activation of two unrelated concepts could be an artifact of the model considered so far.

In order to check whether our findings in Figs 1 and 2 are generic, we added to the network the effect of inhibitory neurons by implementing a negative feedback proportional to the overall activity of the $N$ neurons in the network. Inhibitory feedback of strength $J_0 > 0$ causes competition during recall of memories. We find that for $J_0 = 0.5$, each of the two concepts can be recalled individually, but simultaneous recall of both concepts is not possible if the fraction of shared concept cells is at chance level (Fig 3A). If we increase the fraction of shared concept cells above $c = 5\%$, then individual as well as simultaneous recall of the two associated memories becomes possible (Fig 3B). The effect becomes even more pronounced at $c = 20\%$ (Fig 3C). If the fraction of shared neurons reaches a high value of $c_{max} = 50\%$, then the separate retrieval of the two individual concepts is no longer possible, indicating that the two concepts have merged into a single one (Fig 3D). Thus, in the presence of inhibition of strength $J_0$, we find that the fraction $c$ of shared neurons must be in a range $c_{min}(J_0) < c < c_{max}(J_0)$ to enable individual as well as joint recall of associated concepts. For $J_0 < 0.5$, the minimal fraction $c_{min}(J_0)$ is at chance level and for $J_0 = 0.5$ at $c_{min} = 5\%$.

## Association chains

Neurons shared between memory engrams have been proposed to be the basis for the recall of a memorized list of words [9–12, 26]. In order to translate this idea to chains of associated concepts (Fig 4A), we follow earlier work [9–12, 26] and add two ingredients to the model of the previous subsection. First, the strength of global inhibitory feedback is now periodically modulated by oscillations mimicking Hippocampal oscillatory activity. The oscillations provide a clock signal that triggers transitions between overlapping concepts. Second, we add to each

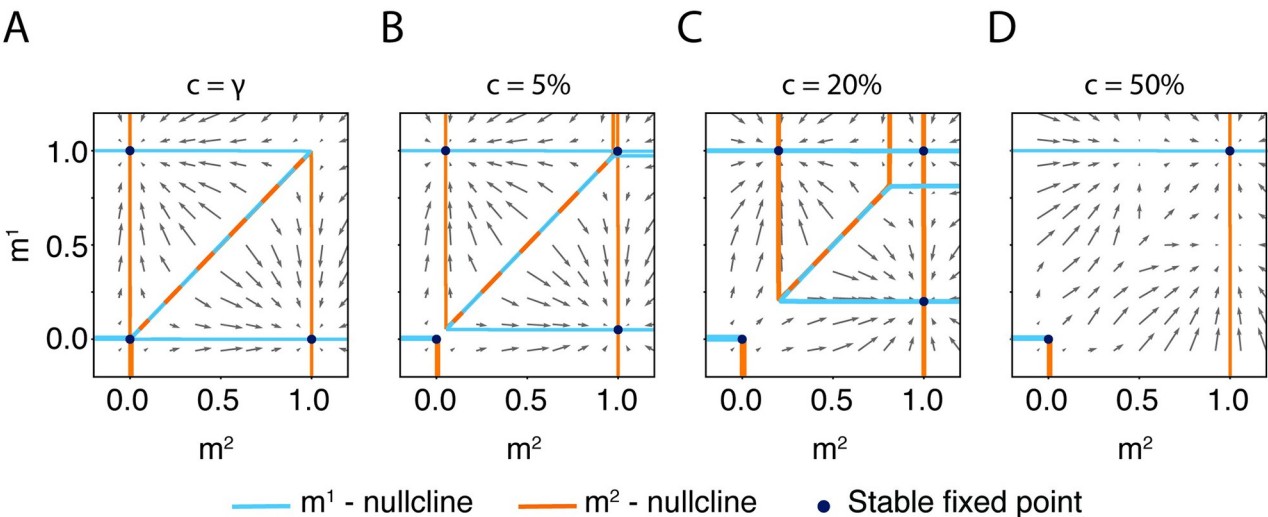

**Fig 3. The existence of a symmetric double-retrieval state requires a fraction of shared neurons above chance level in the presence of global inhibition.** Four phase-planes showing the stable fixed points in presence of global inhibition, for a fractions of shared neurons **A** $c = \gamma$, **B** $c = 5\%$, **C** $c = 20\%$, **D** $c = 50\%$. On the diagonal, nullclines lie nearly on top of each other (dashed line). Parameters: $\hat{h}_0 = 0$, $\hat{b} = 500$, $\alpha = 0$, $\gamma = 0.2\%$, $J_0 = 0.5$.

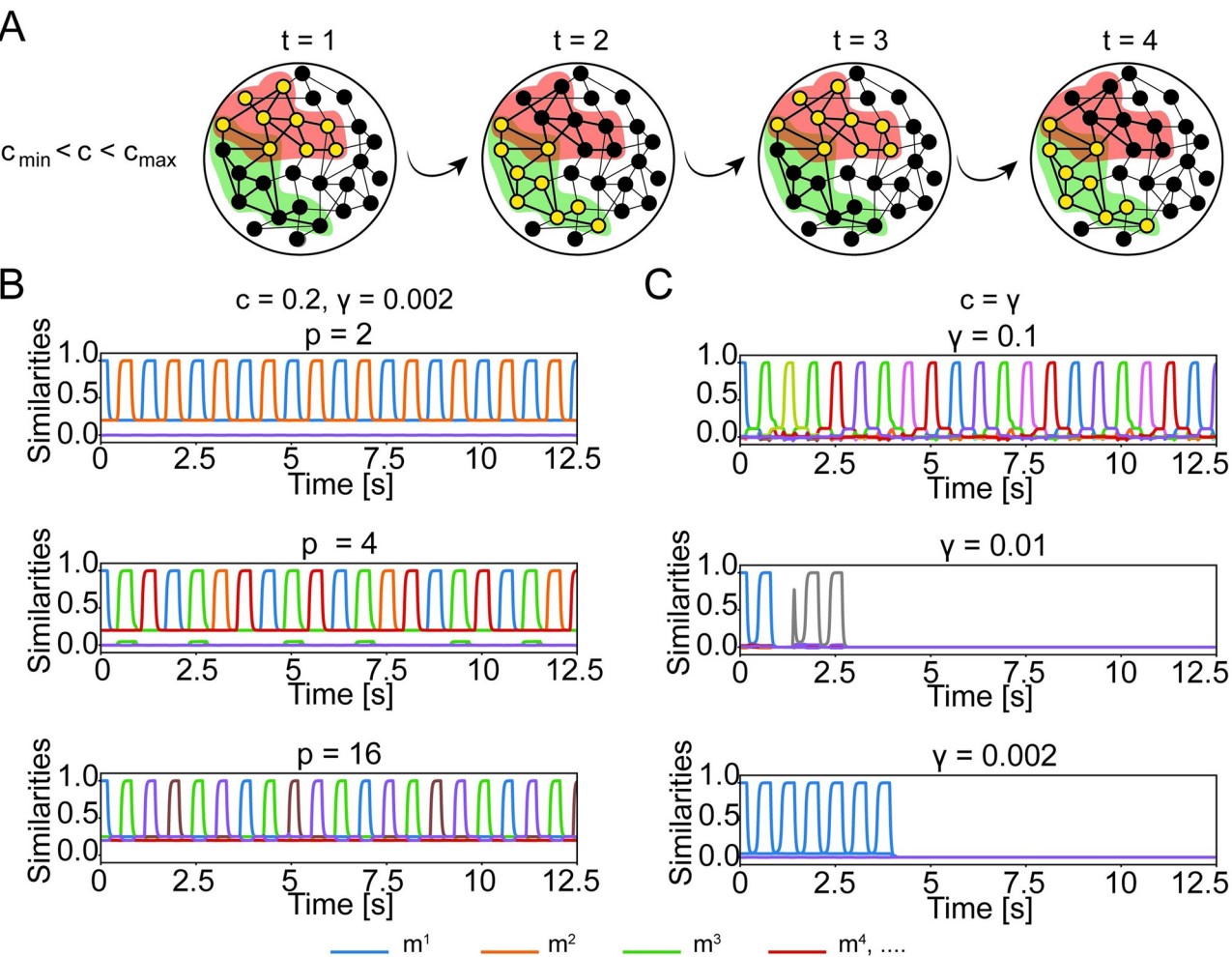

**Fig 4. Chain of associations requires shared concepts cells.** A) Schematic of a chain of association cycling between two concepts. Assignment of cells to assemblies is indicated by halos' color. Filled black circles indicate inactive neurons and filled yellow circles indicate active neurons. The schematics corresponds to the top plot of panel B. B) Full network simulation for engrams overlapping above chance level ($c = 20\% > \gamma$) with low sparsity ($\gamma = 0.2\%$). Each line corresponds to the similarity $m^{\mu}$ with one of the stored memory engrams as a function of time. A subgroup of $p$ engrams is overlapping (top to bottom: $p = 2, 4, 16$. If the network state is initialized to retrieve one of the overlapping concepts, other concepts within the subgroup are retrieved later. C) Same as in B, but memory engrams are independent ($c = \gamma$) and only share cells by chance. By decreasing the mean activity $\gamma$, the retrieval dynamics of a chain of memories is disrupted. The match between mean-field theory and simulations is shown in S4 Fig. Parameters: $N = 10000$, $P = 16$, $\hat{b} \rightarrow \infty$, $\tau_{\theta} = 1.125$ s, $T = 3.75$ ms, $T_{J_0} = 625$ ms, $\tau = 25$ ms, $r_{max} = 40$ Hz.

neuron $i$ an adaptation current $\theta_i(t)$ in order to prevent the network state to immediately return to the previous concept. With this extended model, the network state hops from one concept to the next (Fig 4B). Transitions are repeated, but after some time the network state returns to one of the already retrieved memories, leading to a cycle of patterns [9] (Fig 4B). In network simulations where concepts are represented by sparse memory engrams ($\gamma = 0.2\%$), we allow a subgroup of $p = 2, 4$ or $16$ memory engrams to share a fraction of neurons of $c = 20\%$. Because the number of shared concept cells is identical between all pairs of concepts within the same subgroup, the order of the recalled concepts depends on the initial condition. If the subgroup of overlapping engrams is small ($p = 2, 4$), all memory items are retrieved, while for a large group of overlapping engrams ($p = 16$) the cycle closes once a subgroup of the overlapping memory engrams has been retrieved (Fig 4B). The number of concepts in the

cycle depends on the time scale of adaptation: in Fig 4 we use $\tau_\theta \sim 2T_{J_0}$, which determines a cycle of minimum three concepts.

In previous studies [9–12, 26], each memory engram involved a large fraction ($\gamma = 10\%$) of neurons so that transitions could rely on the number of units shared by chance. The overlaps between memory assemblies may vary due to finite size effects. The concept that shares the biggest overlap with the active one, is activated next, until the association chain falls into periodic cycle of patterns. However, given that the value of the sparsity in MTL is much smaller ($\gamma \sim 0.23\%$), it is natural to ask whether the number of neurons shared by chance ($c = \gamma$) is sufficient to induce a sequence of memory retrievals. Our simulations indicate that this is not the case (Fig 4C). Thus, in a network storing assemblies with a realistic level of sparsity $\gamma \sim 0.2\%$, memory engrams with a fraction of shared neurons above chance level are necessary for the retrieval of chains of concepts.

To better understand the role of overlaps between engrams for the formation of association chains, we extend the mean-field dynamics to include the global feedback with periodic modulation $J_0(t)$. Since simulations indicate that overlaps are necessary, we want to estimate the minimal and maximal fractions of shared neurons required to enable association chains. Because, in our model, the periodic modulation of the global inhibition strength $J_0(t)$ is slow, we consider the mean-field dynamics and the corresponding phase portraits quasi-statically at the two extreme cases, where $J_0$ is at its maximum and where $J_0$ is at its minimum. For our parameter setting, when $J_0(t)$ is clamped at its *minimum*, the network possesses three stable states: the resting state and the two single retrieval states (Fig 5B, left). For a successful association chain, we need that concepts can be retrieved separately. The fraction of shared neurons, $c'_{max}$, that makes the two single retrieval states disappear therefore sets the upper bound of the useful range of $c$. The parameter $c'_{max}$ is analogous to $c_{max}$ in the previous section, but evaluated in the presence of perdiodic inhibition.

Next, we consider the situation when the global inhibition is clamped at its *maximum* and find the minimal fraction such that the system has, besides the resting state, a second fixed point for $m^1 = m^2 > 0$ where the assemblies of both the previous and next concept are simultaneously active at low firing rates. Since this state is necessary to enable the transition, we call it the *transition state*. If the transition state is present, the network could, once global inhibition decreases, either return from the transition state to the previous concept, or jump to the next one (Fig 5B, right side). However, in the presence of adaptation (which is not included in the phase plane picture of Fig 5), the transition to the next concept is systematically favored because neurons participating in the assembly of the earlier concept are fatigued. The existence of the transition state is a necessary condition for the formation of temporal association chains. Thus, the lower bound of the fraction of shared neurons $c'_{min}$ is the smallest overlap such that the transition state exists. Since in the mean-field limit, the transition state appears only for $c > \gamma$, a fraction of shared neurons above chance level is needed to allow the hopping between concepts. In Fig 5C–5E we show the dependence of the maximal and minimal fraction of shared concept cells upon the sparsity $\gamma$ and the steepness $b$: in both cases the dependence is not strong, but sparser networks lead to a slightly smaller range of the admissible fraction $c$ of shared neurons supporting association chains. Importantly, the minimal fraction of shared neurons necessary for association chains is significantly above the fraction of neurons that are shared by chance. We find that for a suitable choice of neuronal and network parameters, association chains are possible for realistic values of $\gamma$ and $c$ as measured in human MTL. This suggests that, in principle, associations could be implemented as sequences of transitions if the number of shared neurons is above $c_{min}$.

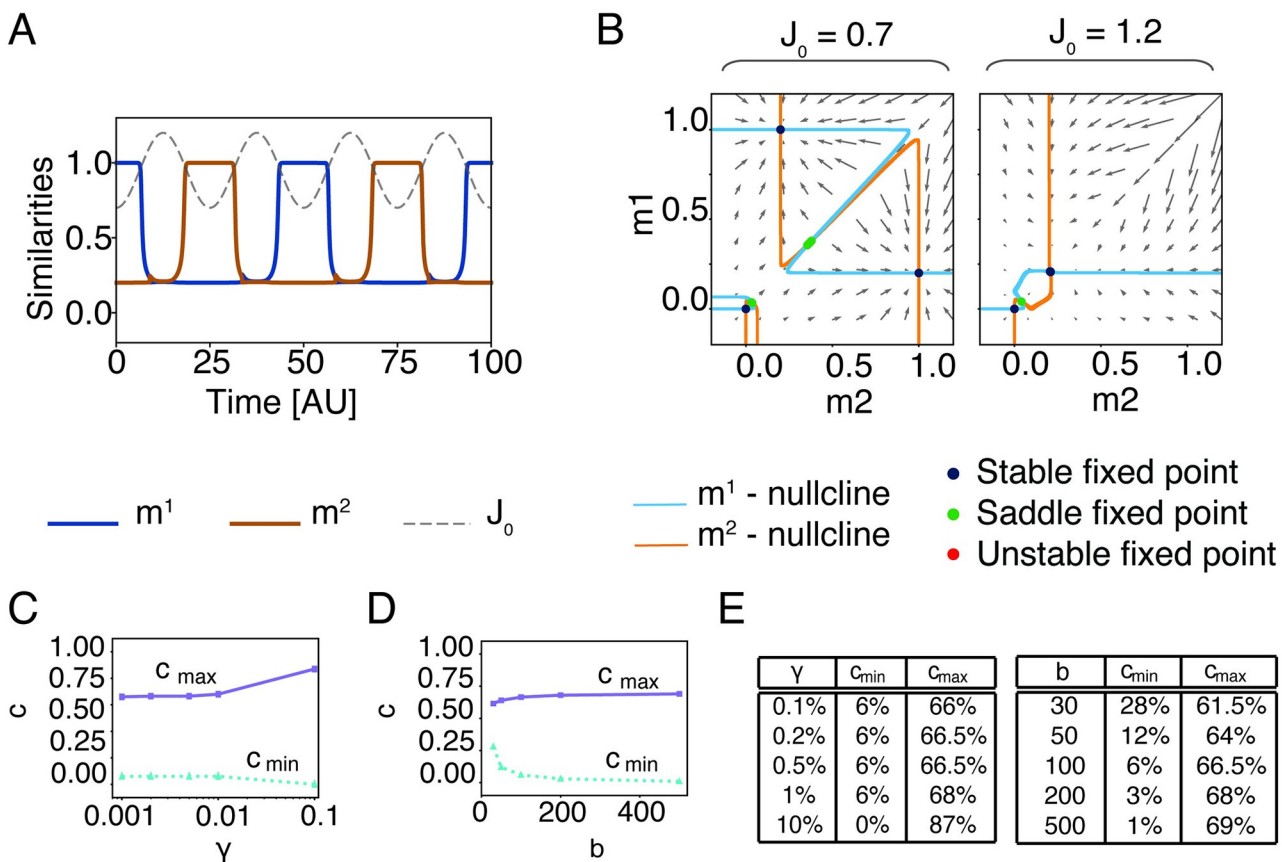

**Fig 5. Dependence of association chains on sparsity and neuronal parameters.** A) Dynamical mean-field solutions for $m^1$ and $m^2$ in the case of two correlated patterns. The grey dashed line shows the modulation of $J_0(t)$. B) Phase planes corresponding to the minimum ($J_0 = 0.7$) and maximum ($J_0 = 1.2$) value of inhibition in the case of two associated patterns. C) Minimal and maximal fraction of shared concept cells as a function of the sparsity $\gamma$ and D) of the steepness $b$. E) Table with the values of C and D. Parameters (unless specified): $\gamma = 0.2\%$, $\hat{b} = 100$, $c = 20\%$, $\tau_\theta = 1.125$ s, $T = 3.75$ ms, $T_{J_0} = 625$ ms, $\theta_i = 0$ for every $i$.

In conclusion, we have shown the need for overlaps between memory engrams—equivalent to a number of shared concept cells significantly above chance level – to explain free memory recall as a chain of associations in recurrent networks such as the human CA3 where each engram involves only a small fraction of neurons.

## How does a network embed groups of overlapping memories?

In our discussion on shared concept cells, we have so far mainly focused on neurons that are shared between a single *pair* of memory engrams such as one place and one person. However, humans are able to memorize many different persons and places, some memories forming subgroups of associated items, others not. In order to compare our network model with human data we therefore need to encode *several* subgroups of *two or more* of overlapping memory engrams in the same network of $N$ neurons. Based on the results of the previous sections, we wondered whether we can explain the experimental distribution of the number of concepts a single neuron responds to. We find that imposing the fraction $c$ of shared concept cells between *pairs* of concepts, does not predict uniquely how many neurons are used if a given number of memory engrams is embedded in a network. Therefore, imposing $c$ as a target

number of shared concept cells while encoding multiple concepts is not sufficient to predict whether a given neuron responds to 3 or 5 different concepts. The question then is: to how many concepts does a single neuron respond if several groups of overlapping engrams have been embedded in the network?.

To study this question we consider three different algorithms that all construct memory engrams of 200 neurons per memory with a pairwise overlap of 8 neurons in a network of 100,000 neurons, i.e., $\gamma = 0.2\%$ and $c = 4\%$ (Fig 6B). First we consider an iterative overlap-generating a non-hierarchical model, in which we impose a fixed target number of shared concept cells as the only condition. Second, we consider two hierarchical models, in which every subgroup of associated memory patterns is derived by a single "parent" pattern, which does not take part in the subgroup. In the hierarchical generative model, only neurons that belong to the parent pattern can be contribute to the neural representation of the patterns in the subgroup. On the contrary, in the indicator neuron model, the parent pattern is composed of indicator neurons that have a fixed probability $\lambda_{ind}$ of appearing in each of the subgroup's patterns. Non-indicator neurons can also take part in the representation of the subgroup's patterns with a different probability $\Omega$. In other words, in the hierarchical generative model, neurons that do not belong to the parent pattern are excluded from the representation of all of the associated patterns in a subgroup, while in the indicator neuron model, no neuron is excluded from contributing to the representation of the subgroup. When we embed subgroups of 16 engrams with identical numbers of pairwise shared neurons, then an iterative overlap-generating model needs about 2400 neurons out of the 100,000 available neurons, whereas two different hierarchically organized algorithms need about 2400 or 3000 neurons, respectively. In order to understand which of the three algorithms explains experimental data best, we quantify the predictions of the three algorithms under the assumption that not just one, but several subgroups of patterns are embedded in the same network and compare the predictions with experimental data using a previously published dataset of human concept cells [6].

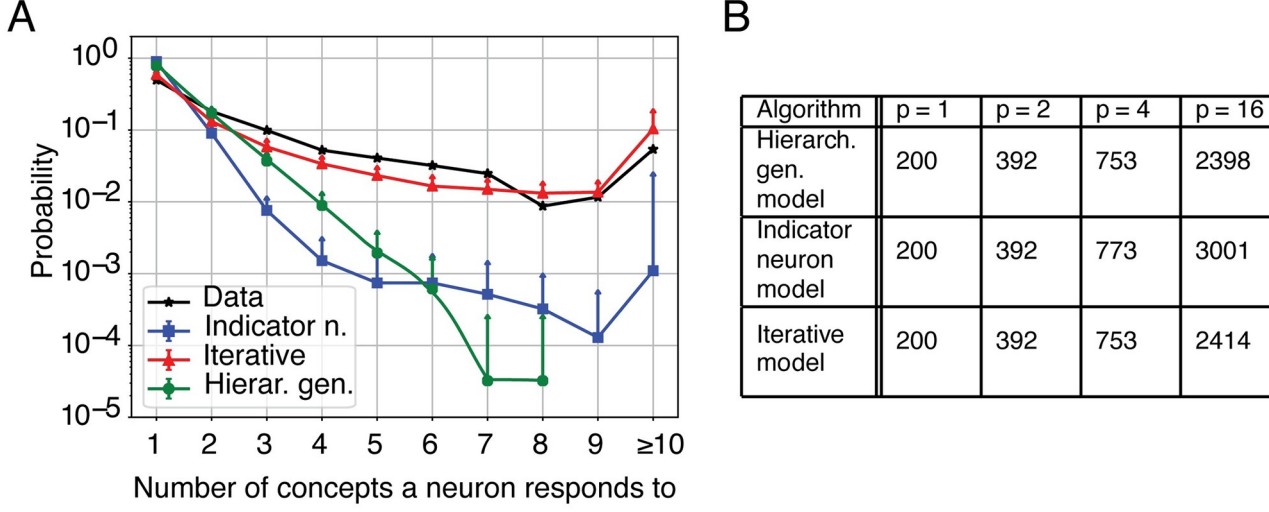

**Fig 6. A single neuron responds to several concepts.** A) Probability that a neuron responds to a given number of concepts: comparison between data and 3 different algorithms: the hierarchical generative model and the indicator neuron model, which both build overlapping engrams in a hierarchical way, and the iterative overlap-generating model which is a non-hierarchical algorithm. Each algorithm was run 40 times to generate the mean and error bars (only upward bars are displayed, corresponding to one standard deviation). B) For each of the three algorithms, we generated three subgroups of patterns containing $p = 16$, $p = 4$, or $p = 2$ patterns, respectivley, as well as an isolated pattern ($p = 1$). The table gives the expected total number of active neurons in each subgroup in a neural network of 100000 neurons if patterns have sparsity $\gamma = 0.2\%$ and a pairwise fraction of shared neurons $c = 4\%$.

The dataset contains the activity of 4066 neurons recorded from the human MTL during the presentation of several visual stimuli. We can extract the experimental probability that a single neuron responds to exactly $k$ different concepts (Fig 6A, black stars). From the probability distribution, we observe the existence of neurons responding to a large number of concepts (10 or more), but also a sizable fraction of neurons that respond to 5 or 6 different concepts. We will refer to those neurons as multi-responsive neurons.

To describe the data, we take into account the size and number of subgroups used in the experimental stimulation paradigm (SI). We find that only the iterative overlap-generating model fits the data (Fig 5C), i.e., it is the only one that predicts the correct probability of multi-responsive neurons. Since the iterative overlap-generating model is not based on a hierarchical generation of patterns, this suggests that the MTL encodes large subgroups of memory engrams in a non-hierarchical way, in agreement with earlier papers [6].

## Robustness to heterogeneity

Because biological neural networks present different forms of heterogeneity, we have checked our model's robustness to (i) the heterogeneity of frequency-current curves and (ii) dilution of the number of synaptic connections.

In the experimental data set, each neuron is characterized by different baseline firing rates and maximal rates in response to the preferred stimulus. We therefore introduce in our model heterogeneous frequency-current curves characterised by a minimum and a maximum firing rates $(r_{min})_i$ and $(r_{max})_i$ respectively and renormalize the network dynamics appropriately (Materials and methods). Despite the heterogeneity, simulations indicate that memory recall with heterogeneity is nearly indistinguishable from that without (compare Figs 7A and 2C and 7B and 4B).

Secondly, we allow the weight matrix to be diluted. Whereas so far we have assumed an "all-to-all" connectivity, we now introduce the dilution coefficient $d$, which indicates the fraction of actual synaptic connections compared to the $N^2$ potential ones. Importantly, for sparsely connected networks, the theory still contains the parameter $\alpha$ for memory load, except that $\alpha$ is redefined to $\alpha = P/M$ where $M$ is the mean number of connections per neuron (Materials and methods). Simulations in Fig 7C show that the model is robust for $d = 0.8$, i.e. after dropping 20% of all possible synaptic connections and an appropriate rescaling of the average connection strength. We have explored lower values of connection probability $d$, to approach a more bio-plausible regime [27]. However, increasing the dilution of the connections takes the dull network simulations away from the mean-field regime in which the theory is valid. The problem could be overcame by increasing proportionally the size of the network size $N$, but the computational cost grows with the square of network size $N$. The optimization of the simulation of a very diluted attractor neural network is beyond the goals of the current work.

## Discussion

Our results bridge observations and theories from four different fields: first, experimental observations in the human MTL [1, 3, 6, 28, 29]; second, experimental observations of memory engrams [18, 19]; third, the theory of association chains used to explain free memory recall [9–12]; and fourth the classic theory of attractor neural networks [13, 25]. Our main result is that, in networks where concepts are encoded by sparse assemblies, the number of shared concept cells must be above chance level but below a maximal number in order to enable a reliable encoding of associations. With 4–5% overlap between memory assemblies as reported in the human MTL [6], association chains are possible for a range of parameters of frequency-current

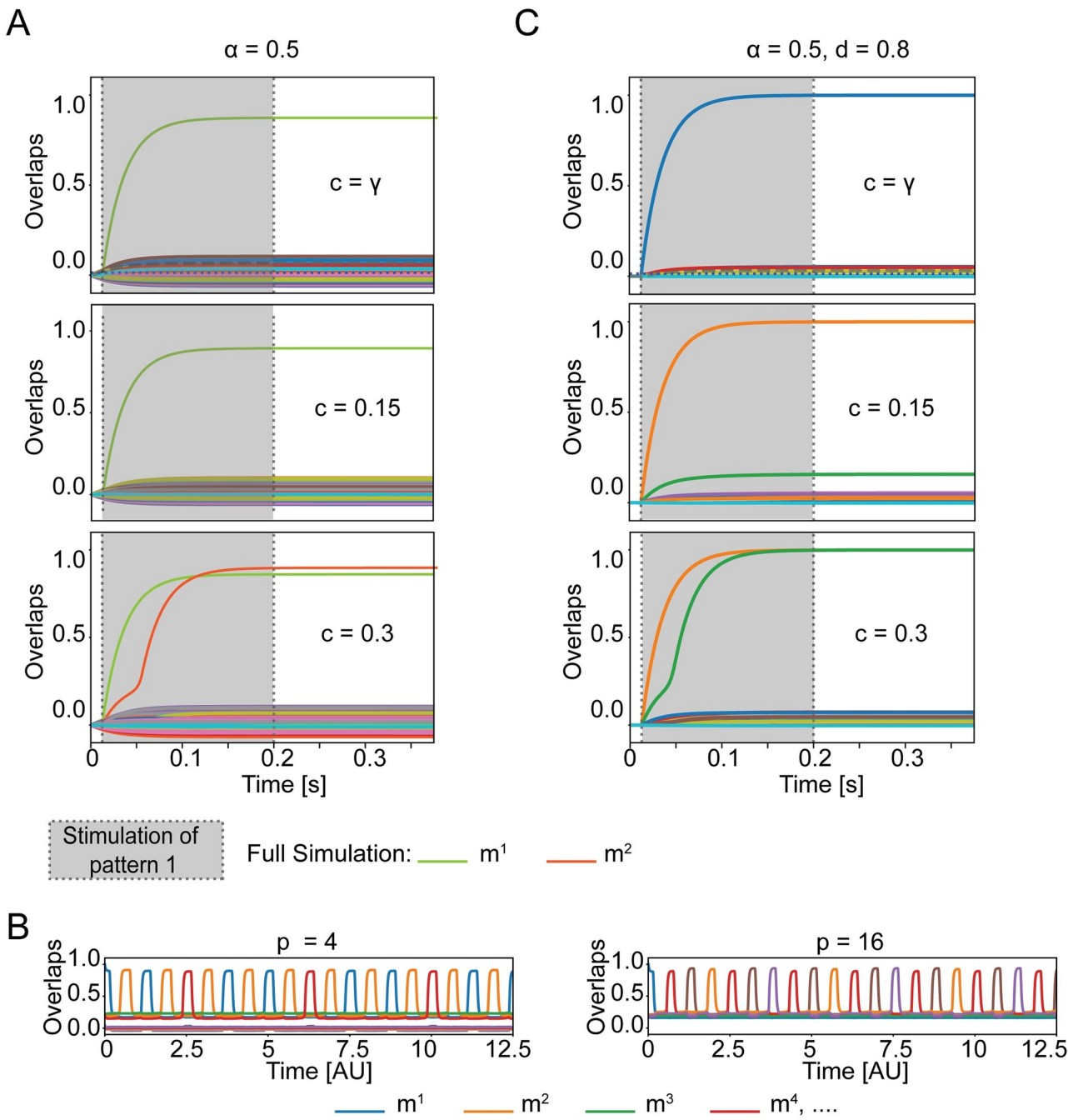

**Fig 7. The model is robust to heterogeneity of frequency-current curves.** Full network simulations A) in absence of adaptation (equivalent to Fig 2C) and B) in presence of adaptation and periodic inhibition. C) The model is robust to the dilutions of the synaptic connections. Full network simulations equivalent to Fig 2C.

curves. Our work extends the classical mean-field formalism [15] to memory engrams that exhibit pairwise overlap, both in a static and chain-like retrieval setting.

While sparsity limits the number of concept cells shared by chance, Hebbian learning could induce sharing of concept cells between a small number of specific memories engrams [6]. The

existence of a maximal fraction of shared neurons implies that Hebbian learning must work with an intrinsic control mechanism so as to avoid unwanted merging of separate concepts.

Our model allows us to make novel predictions for experiments. We see from Fig 1C and 1D that stimulating an associated concept is easier in the presence of shared concept cells than without. We extend this paradigm to several concepts and form the following prediction. Imagine having two sub-groups of overlapping memories both involving the same person P0, in one case related to a person P1 and in another case to a person P2. How can we dissociate between the 2 memories (P0 and P1 vs P0 and P2)? Our model predicts that the dissociation is possible by introducing different contexts (e.g., different places). Say P0 and P1 would be related to context C1 (Barcelona) and P0 and P2 would be related to context C2 (Pisa), so whenever P0 is activated together with C1, P1 will also tend to be activated but not P2 and whenever P0 is activated in context C2, P2 will tend to be co-activated but not P1. In Fig 8 we illustrate the experimental setup with the following simulation: we store 5 concepts, two contexts C1 and C2 and three persons P0, P1, P2. The context could be place C1 that has quite strong overlap with both concepts of persons P0 and P1, but not P2. The place C2, instead has overlap with P0 and P2 but not P1. During the simulation we give a weak ambiguous stimulation to P1 and P2. If we stimulate the concept of person P0, P0 has overlap with P1 and P2 and so far we have no bias in either way. Later we activate context C1, and this disentangles memories by favouring the recall of P0 together with P1. We emphasize that the activation of C1 provides a bias to activate concepts of Persons P0 and P1 but this bias is not strong enough on its own to recall P0 or P1. Even the co-activation of C1 and P0, is not enough, to automatically activate P1 on its own, in the context of the static model without adaptation, or inhibition. In our framework the activation of one context favors the recall of concepts associated to it, and it can be qualitative compared to neuron-specific gating model proposed in [30], where the activation of one context defines a subset of available neurons.

Association chains could form the basis of a "stream of thought" where the direction of transitions from one concept to the next is based on learned associations. Our oscillatory network dynamics is inspired by the model of Romani, Tsodyks and collaborators [9–12]. Even though in the Romani-Tsodyks model memory engrams are independent, finite size effects make some pairs of engrams share neurons above chance level which enables sequential recall

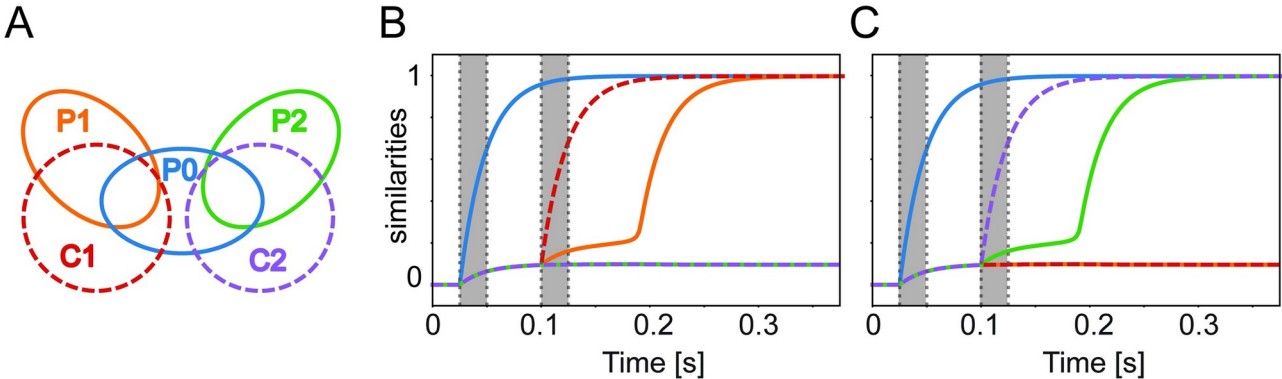

**Fig 8. Simulation procedure that predicts that the context disentangles memories.** A) schematics of the overlaps between the five stored concepts: three persons (P0, blue; P1, orange; P2, green) and two contexts (C1, red; C2, violet) B) During the first stimulation period (indicated by the shaded grey area) the concept of Person 0 (P0) is strongly stimulated and during the second grey period, the concept of Context 1 (C1) is strongly stimulated, leading to the activation of P1. Person 1 (P1) and 2 (P2) are always weakly stimulated. C) Same as B, but in the second stimulation period we strongly stimulate concept C2 instead of C1, leading to the activation of P2. Parameters: intensity of the weak stimulation = 0.02, intensity of the strong stimulation = $r_{max}$, $N = 20000$, $P = 5$, $\hat{b} = 100$, $\tau = 25$ ms, $r_{max} = 40$ Hz, correlation within each of the two sub-groups P0-P1-C1 and P0-P2-C2, $C = 0.1$.

in the presence of a periodic background input. We find that in large networks with sparse coding level ($\gamma \approx 0.23\%$), neurons shared by chance are not enough to reliably induce the retrieval of a chain of concepts. Sequential memory retrieval is possible only for overlaps larger than chance, potentially representing associations learned during real-life episodes. Instead of transitions triggered by oscillations, transitions could also be triggered by two adaptation mechanisms that act on different time scales without the need of periodic inhibition [31–33].

Attractor networks with sparse patterns [17] and random connectivity [34] are suitable candidate models for biological memory because they present two features: (i) memory retrieval after stimulation with a partial cue and (ii) sustained activity after a stimulus has been removed. One of the points of critique of attractor networks, traditionally analyzed with the replica [35] or cavity [36, 37] method, has been the unrealistic assumption of symmetric connections. However, the derivation used here, based on dynamical systems arguments [38], can easily be generalized to the case of asymmetric connectivity.

We discuss the possibility of allowing all patterns to share the same amount of correlation at the very end of section "Overlapping background patterns", in the MATERIALS AND METHODS. In this case, we show that the standard deviation of the background noise is proportional to $P^2/N$. If we make the standard mean-field assumption that both P and N tend to infinity with constant ratio $P/N = \alpha$, then the quenched noise due to the presence of background patterns would diverge. However, we can define the memory load $\alpha' = P^2/N$ and we can assume that P and N tend to infinity, keeping $\alpha'$ constant. In this scenario, the network capacity is drastically reduced. Alternative approaches to keep into account the fact that some neurons are more easily recruited, can be 1. To assign heterogeneous gain functions, therefore making some neurons more excitable, 2. To consider those neurons to be the "multi-responsive neurons" that we describe in section "How does a network embed groups of overlapping memories".

The maximal number of patterns that can be stored in an attractor neural networks has attracted a lot of research [15, 17, 39]. However, does the hippocampus actually operate in the regime of high memory load? Even though we do not believe that hippocampus stores words, we may estimate a rough upper bound for the load $\alpha = P/M$ in the area CA3 of the hippocampus from the number of words a native English speaker knows (which is about $P = 30'000$ according to The Economist, Lexical Facts) and the number of input connections per neuron (which is about $M = 30'000$ [40]). Hence we estimate an upper bound of $\alpha$ about 1 if concepts are stored in area CA3—and our theory captures such a high load.

The maximal fraction of neurons which two concepts can share before they effectively merge into a single concept mainly depends on two dimensionless parameters: the rescaled threshold $\hat{h}_0 = h_0/(Ar_{\max})$ and the rescaled steepness $\hat{b} = Ar_{\max}b$. Since these parameters have so far not been estimated for the human CA3 area of the hippocampus or for the MTL in general, we checked parameters of the frequency-current curve of Macaque inferotemporal cortex [24], for which we find $c_{\max} = 34\%$.

Finally, by comparing the experimental measured number of concepts a neurons responds to and model predictions we find that the iterative overlap-generating model can predict the number of multi-responsive neurons quite accurately. The algorithm of how to build overlapping engrams plays a key role in fitting the experimental data and confirms the idea that memory engrams in the hippocampus are not hierarchically organised.

## Materials and methods

We consider an attractor neural network of N rate units with firing rates $r_i$, $i = 1, \ldots, N$, in which P memory engrams are stored. Each engram $\mu$, $1 \leq \mu \leq P$, is given by a binary random pattern $\vec{x}^\mu = [\xi_1^\mu, \ldots, \xi_N^\mu]^T$, where $\xi_i^\mu \in \{0, 1\}$ are Bernoulli random variables with mean

$\langle \xi_i^\mu \rangle = \gamma$. Here and in the following, $\langle . \rangle$ indicates expectation over the random numbers $\xi_i^\mu$ that make up the patterns. Each neuron follows the rate dynamics of Eq (1), where the synaptic weight from neurons $j$ to neurons $i$ is defined as [17, 24]

$$w_{ij} = \frac{A}{N\gamma(1-\gamma)} \sum_{\mu=1}^{P} \left(\xi_j^\mu - \gamma\right)\left(\xi_i^\mu - \gamma\right). \tag{5}$$

Here, the constant $A$ can be interpreted as the global scale of "connection strength". For independent patterns, the synaptic weight $w_{ij}$ has mean zero, $\langle w_{ij} \rangle = 0$, and variance $\langle w_{ij}^2 \rangle = A^2 P/N^2$.

## Model without adaptation and global feedback

In the results in Figs 1 and 2, each neuron follows the Wilson-Cowan dynamics [41]

$$\tau \frac{dr_i}{dt} = -r_i + \phi(h_i), \tag{6}$$

where the total input driving neuron $i$ is

$$h_i(t) = \sum_{j=1}^{N} w_{ij} r_j(t) + I_i(t) = \sum_{\mu=1}^{P} (\xi_i^\mu - \gamma) m^\mu(t) + I_i(t). \tag{7}$$

Here, $I_i$ is the external input to neuron $i$ and

$$m^\mu(t) = \frac{1}{N\gamma(1-\gamma)r_{\max}} \sum_{j=1}^{N} \left(\xi_j^\mu - \gamma\right) r_j(t). \tag{8}$$

is the similarity measure (also called "overlap" in the attractor network literature). It measures the similarity (correlation) of the current network state with pattern $\mu$; cf. Eq (2). In Figs 1C (during the first stimulation period), 2C and 6 the external input $I_i = I^{\text{ext}}\xi_i^1$ is positive during stimulation for all neurons that belong to the assembly of pattern $\mu = 1$, and zero for all other neurons.

In Eq (6), the input is passed through the transfer function $\phi$ (also called f-I curve in the Results section), which is chosen to be a sigmoid:

$$\phi(h) = \frac{r_{\max}}{1 + e^{-b(h-h_0)}}. \tag{9}$$

The parameters that define the transfer function can be interpreted as follows: $r_{\max}$ is the maximal firing rate, $b$ is the steepness of the transfer function and $h_0$ is the bias which is commonly interpreted as firing threshold. While $h_0$ is a hard threshold for $b \to \infty$, at finite $b$ the model exhibits a soft threshold allowing firing activity even below $h_0$.

## Model with adaptation and global inhibitory feedback

For Fig 4 of Results, we added adaptation and a global inhibitory feedback to the model as described in previous studies [9–12] (see also [42] for a similar rate model with adaptation). Specifically, we add two negative feedback terms to the input potential:

$$h_i(t) = \sum_{j=1}^{N} w_{ij} r_j(t) - \theta_i(t) - \frac{J_0(t)}{\gamma N} \sum_{j=1}^{N} r_j(t) + I_i, \tag{10}$$

First, the variable $\theta_i(t)$ models neuron-specific firing-rate adaptation via the first-order kinetics

$$\tau_\theta \frac{d\theta_i}{dt} = -\theta_i + D_\theta r_i. \tag{11}$$

Here, $\tau_\theta$ is the adaptation time constant and $D_\theta$ determines the strength of adaptation. Note that this adaptation model with a hyperpolarizing feedback current is equivalent to a model in which adaptation is implemented as an increase in the threshold $h_0 + \theta_i(t)$.

Second, the global inhibitory feedback term proportional to $J_0(t)$ (third term in Eq (10)) provides a clock signal that triggers transitions between attractors. The strength of the global feedback, $J_0(t)$, is modulated periodically in time:

$$J_0 = \frac{1}{2}(J_{\max} - J_{\min})\sin\left(\frac{2\pi}{T_{J_0}}t - \frac{\pi}{2}\right) + \frac{1}{2}(J_{\max} + J_{\min}) \tag{12}$$

Importantly, inhibition proportional to the summed activity of the network units penalizes network configurations with many active neurons and therefore reduces stability of the double-retrieval state where two memories are recalled together. Here, the strength $J_0(t)$ of the global feedback is modulated periodically between values 0.7 and 1.2 with a sinusoidal time course of period $T_{J_0}$ that sets the time scale of transitions between memories. Note that the model without adaptation and global feedback is a special case of the full model by setting $D_\theta = 0$ and $J_0(t) \equiv 0$. For the results of Fig 3, $J_0$ is a constant parameter and $D_\theta = 0$.

## Non-dimensionalization of the model

The calculations below are considerably simplified if the model is nondimensionalized. We take into account that $r_{\max}$ has units of 1/time and the parameter $A$ has units of current · time and measure in the following time in units of $r_{max}^{-1}$ and current input in units of $Ar_{max}$.

**Model without adaptation and global feedback.** Using the dimensionless quantities

$$\hat{h}_i = \frac{h_i}{Ar_{\max}}, \quad \hat{h}_0 = \frac{h_0}{Ar_{\max}}, \quad \hat{b} = bAr_{\max}, \quad \hat{r}_i = \frac{r_i}{r_{\max}} \tag{13}$$

$$\hat{w}_{ij} = \frac{w_{ij}}{A}, \quad \hat{\tau} = \tau r_{\max}, \quad \hat{I}_i(t) = \frac{I_i(t)}{Ar_{\max}}, \tag{14}$$

the nondimensionalized model without adaptation reads

$$\hat{\tau}\frac{d\hat{r}_i}{d\hat{t}} = -\hat{r}_i + \hat{\phi}(\hat{h}_i), \text{ with } \hat{h}_i = \sum_{j=1}^{N}\hat{w}_{ij}\hat{r}_j + \hat{I}_i \tag{15}$$

with a transfer function $\hat{\phi}(\hat{h}) = 1/\{1 + \exp[-\hat{b}(\hat{h} - \hat{h}_0)]\}$.

**Model with adaptation and global feedback.** Introduction of further dimensionless quantities

$$\hat{\tau}_\theta = \tau_\theta r_{\max}, \quad \hat{\theta} = \frac{\theta}{Ar_{max}}, \quad \hat{D}_\theta = \frac{D_\theta}{A}, \quad \hat{J}_0(\hat{t}) = \frac{J_0(\hat{t}/r_{\max})}{A} \tag{16}$$

leads to the nondimensionalized model with adaptation

$$\hat{\tau}\frac{d\hat{r}_i}{d\hat{t}} = -\hat{r}_i + \hat{\phi}(\hat{h}_i),$$  (17)

$$\hat{\tau}_\theta \frac{d\hat{\theta}_i}{d\hat{t}} = -\hat{\theta}_i + \hat{D}_\theta \hat{r}_i$$  (18)

with input

$$\hat{h}_i = \sum_{j=1}^N \hat{w}_{ij}\hat{r}_j - \hat{\theta}_i - \frac{\hat{J}_0(t)}{\gamma N}\sum_{j=1}^N \hat{r}_j(t) + \hat{I}_i.$$  (19)

## Review of attractor theory

Starting from the overlap definition Eq (2), we can write equations for the overlaps variables. We first focus on the model without adaptation and global feedback. For this case, we follow an approach well-known in literature [17, 38]. Taking the temporal derivative of the similarity $m^\mu$ yields

$$\hat{\tau}\frac{dm^\mu}{d\hat{t}} = \frac{1}{N\gamma(1-\gamma)}\sum_{j=1}^N \left(\xi_j^\mu - \gamma\right)\hat{\tau}\frac{d\hat{r}_j}{d\hat{t}}.$$  (20)

By inserting the expression for the single neuron dynamics Eq (15) and recognizing the overlap definition Eq (2), we obtain:

$$\hat{\tau}\frac{dm^\mu}{d\hat{t}} = -m^\mu + F_\mu(m^1, \ldots, m^P).$$  (21)

with

$$F_\mu(m^1, \ldots, m^P) = \frac{1}{N\gamma(1-\gamma)}\sum_{j=1}^N \left(\xi_j^\mu - \gamma\right)\hat{\phi}\left(\hat{h}_j\right).$$  (22)

In this equation, the dependence on the overlaps $m^1, \ldots, m^P$ is contained in the input term $\hat{h}_j$. From Eq (15) and by using the definition of the weights $w_{ij}$, Eq (5), we have

$$\hat{h}_i = \sum_{\mu=1}^P (\xi_i^\mu - \gamma)m^\mu + \hat{I}_i.$$  (23)

In what follows, we are interested in finding equilibrium solutions of Eq (21), for which $m^\mu = F_\mu(m^1, \ldots, m^P)$. Because we are interested in pattern retrieval, we consider, without loss of generality, the retrieval of pattern 1. To this end, we assume that among all $m^\mu$, only $m^1$ is significantly larger than zero. This network state could be the result of a stimulation in the direction of pattern 1: $\hat{I}_i(t) = \hat{I}(t)\xi_i^1$. Under this assumption we can re-write the input term $\hat{h}_i$ isolating the contribution from $m^1$

$$\hat{h}_i = (\xi_i^1 - \gamma)m^1 + \sum_{\mu=2}^P (\xi_i^\mu - \gamma)m^\mu + \hat{I}(t)\xi_i^1.$$  (24)

We call the patterns that are not recalled "background patterns"; in the present case, these are all patterns for which $\mu \geq 2$. The second term on the r.h.s of Eq 24 represents the

contribution from the background patterns causing some degree of heterogeneity of the input potential for neurons with the same selectivity to pattern 1. For large $P$, this heterogeneity can be captured by replacing the term $\sum_{\mu=2}^{P} (\xi_i^\mu - \gamma) m^\mu$ by a Gaussian random variable with mean zero and variance

$$\sigma^2 = \frac{1}{N} \sum_{i=1}^{N} \sum_{\mu=2}^{P} \sum_{v=2}^{P} \left(\xi_i^\mu - \gamma\right)\left(\xi_i^v - \gamma\right) m^\mu m^v = \gamma(1-\gamma) \sum_{v=2}^{P} (m^v)^2 \qquad (25)$$

To obtain the result in Eq (25), we used the assumption that patterns $\xi_i^\mu$ and $\xi_i^v$ are uncorrelated, and the fact that only the term for $\mu = v$ survives because $\langle(\xi_i^\mu \xi_i^v + \gamma^2 - \gamma\xi_i^v - \gamma\xi_i^\mu)\rangle_i = \delta_{\mu v}\gamma(1-\gamma)$. Here and in the following, the brackets $\langle x_i \rangle_i$ of a variable $x_i$ denotes the population average $\langle x_i \rangle_i = \frac{1}{N}\sum_{i=1}^{N} x_i$. In the next passages, we compute $(m^\mu)^2$, $\mu \neq 1$, in the large network limit $N \to \infty$. For $\mu = 2, \ldots, P$, we expand Eq (22) around $m^\mu = 0$ up to first order in $m^\mu$,

$$F_\mu(m^1, \ldots, m^P) \approx \frac{1}{\gamma(1-\gamma)N} \sum_{j=1}^{N} \left[(\xi_j^\mu - \gamma)\hat{\phi}(\hat{h}_j)|_{m^\mu=0} + (\xi_j^\mu - \gamma)^2 \hat{\phi}'(\hat{h}_j)|_{m^\mu=0} m^\mu\right]. \qquad (26)$$

At equilibrium, $m^\mu = F_\mu(m^1, \ldots, m^P)$, and we thus have

$$m^\mu \left(1 - \sum_{j=1}^{N} \frac{(\xi_j^\mu - \gamma)^2 \hat{\phi}'(\hat{h}_j)}{\gamma(1-\gamma)N}\right) = \frac{1}{\gamma(1-\gamma)N} \sum_{j=1}^{N} \left(\xi_j^\mu - \gamma\right)\hat{\phi}(\hat{h}_j), \qquad \mu \geq 2. \qquad (27)$$

On the left hand side of the last expression, we can make some simplification, utilizing the fact that $\xi_j^\mu$ is uncorrelated with $\phi'(h_j)$ in the $N \to \infty$ limit:

$$\lim_{N\to\infty} \frac{1}{N} \sum_{j=1}^{N} \frac{(\xi_j^\mu - \gamma)^2 \hat{\phi}'(\hat{h}_j)}{\gamma(1-\gamma)} = \lim_{N\to\infty} \frac{1}{N} \sum_{j=1}^{N} \hat{\phi}'(\hat{h}_j) = \langle \hat{\phi}'(\hat{h}_i)\rangle_i. \qquad (28)$$

We can therefore define the quantity $q := \langle\hat{\phi}'(\hat{h}_i)\rangle_i$ as the expectation of $\hat{\phi}'(\hat{h}_i)$ over neurons. As a consequence, $m^\mu$ can be written as

$$m^\mu = \frac{1}{\gamma(1-\gamma)(1-q)N} \sum_{j=1}^{N} \left(\xi_j^\mu - \gamma\right)\hat{\phi}(\hat{h}_j). \qquad (29)$$

Using this equation, we can finally compute the square of $m^v$ for $v \geq 2$:

$$(m^v)^2 = \frac{1}{\gamma^2(1-\gamma)^2(1-q)^2 N^2} \sum_{i=1}^{N} \sum_{j=1}^{N} \left(\xi_i^\mu - \gamma\right)\left(\xi_j^\mu - \gamma\right)\hat{\phi}(\hat{h}_i)\hat{\phi}(\hat{h}_j), \qquad (30)$$

$$= \frac{1}{\gamma^2(1-\gamma)^2(1-q)^2 N^2} \sum_{i=1}^{N} (\xi_i^\mu - \gamma)^2 [\hat{\phi}(\hat{h}_i)]^2, \qquad (31)$$

$$= \frac{p}{\gamma(1-\gamma)(1-q)^2 N}. \qquad (32)$$

where $p := \langle\hat{\phi}^2(\hat{h}_i)\rangle_i$. Similarly as in Eq (25), we used that in the double sum $\sum_{i=1}^{N}\sum_{j=1}^{N}$, only

the term $i = j$ survives:

$$\frac{1}{N^2}\sum_{i=1}^{N}\sum_{j=1}^{N}\left(\xi_i^\mu - \gamma\right)\left(\xi_j^\mu - \gamma\right)\hat{\phi}(\hat{h}_i)\hat{\phi}(\hat{h}_j) = \frac{1}{N^2}\sum_{i=1}^{N}(\xi_i^\mu - \gamma)^2[\hat{\phi}(\hat{h}_i)]^2 +$$

$$\frac{1}{N^2}\sum_{i=1}^{N}\sum_{j\neq i}^{N}\left(\xi_i^\mu \xi_j^\mu - \gamma\xi_i^\mu - \gamma\xi_j^\mu + \gamma^2\right)\hat{\phi}(\hat{h}_i)\hat{\phi}(\hat{h}_j). \tag{33}$$

The population average in the last term factorizes owing to the independence of $\xi_i^\mu$ and $\hat{h}_i$ in the limit $N \to \infty$, and thus vanishes:

$$\frac{1}{N^2}\sum_{i=1}^{N}\sum_{j\neq i}^{N}\left(\xi_i^\mu \xi_j^\mu - \gamma\xi_i^\mu - \gamma\xi_j^\mu + \gamma^2\right)\hat{\phi}(\hat{h}_i)\hat{\phi}(\hat{h}_j) =$$

$$\left[\frac{1}{N^2}\sum_{i=1}^{N}\sum_{j\neq i}^{N}\left(\xi_i^\mu \xi_j^\mu - \gamma\xi_i^\mu - \gamma\xi_j^\mu + \gamma^2\right)\right] \cdot \left[\frac{1}{N^2}\sum_{i=1}^{N}\sum_{j\neq i}^{N}\hat{\phi}(\hat{h}_i)\hat{\phi}(\hat{h}_j)\right] = 0 \tag{34}$$

Here, we have used that the first factor is a vanishing population average: $\gamma^2 - 2\gamma^2 + \gamma^2 = 0$. The standard deviation of the neuron-to-neuron variability (heterogeneity), Eq (25), is thus

$$\sigma = \sqrt{\alpha R}, \qquad R := \frac{p}{(1-q)^2}. \tag{35}$$

As a result, the input potentials, Eq (24), can be expressed at equilibrium as

$$\hat{h}_i = (\xi_i^1 - \gamma)m^1 + \sqrt{\alpha R}Z_i + I \cdot \xi_i^1, \tag{36}$$

where $Z_i \sim N(0, 1)$ are Gaussian random variables. Therefore, we find from Eq (22) that the overlap $m^1$ at equilibrium satisfies

$$m^1 = F_1(m^1) := \frac{1}{\gamma(1-\gamma)}\langle(\xi_i^1 - \gamma)\hat{\phi}(\hat{h}_i)\rangle_i, \tag{37}$$

where $\hat{h}_i$ is given by Eq (36). The population averages $\langle \cdot \rangle_i$ can be treated as expectations over the independent random variables $\xi_i^1$ and $Z_i$. On the one hand, $\xi_i^1$ is a Bernoulli variable such that $\xi_i^1 = 1$ with probability $P_1 = \gamma$ and $\xi_i^1 = 0$ with probability $P_0 = 1 - \gamma$. On the other hand, $Z_i$ is a standard normal random variable with probability density $p_z(z) = \exp(-z^2/2)/\sqrt{2\pi}$. We can therefore rewrite the population average in Eq (37) explicitly resulting in

$$F_1(m^1) = \frac{1}{\gamma(1-\gamma)}\sum_{k=0,1}P_k(k-\gamma)\int \hat{\phi}(\hat{h}_k(m^1, z))e^{-\frac{z^2}{2}}\frac{dz}{\sqrt{2\pi}} \tag{38a}$$

where we defined

$$\hat{h}_k(m, z) = (k - \gamma)m + \sqrt{\alpha R(m)}z + Ik, \qquad k \in \{0, 1\}. \tag{38b}$$

$$R(m) = \frac{p(m)}{[1 - q(m)]^2} \tag{38c}$$

$$q(m) = \sum_{k=0,1}P_k\int \hat{\phi}'(\hat{h}_k(m, z))e^{-\frac{z^2}{2}}\frac{dz}{\sqrt{2\pi}} \tag{38d}$$

$$p(m) = \sum_{k=0,1}P_k\int \hat{\phi}^2(\hat{h}_k(m, z))e^{-\frac{z^2}{2}}\frac{dz}{\sqrt{2\pi}}. \tag{38e}$$

## Dynamical mean-field equations

Approximating the function $F_1(m^1, \ldots, m^P)$ in the dynamical Eq (21) by the simplified function $F_1(m^1)$ derived in the previous section, the retrieval of pattern 1 can be described by the closed dynamical mean-field equation

$$\hat{\tau} \frac{dm^1}{d\hat{t}} = -m^1 + F_1(m^1). \tag{39}$$

For small network load, $\alpha \ll 1$, the effect of background patterns in Eqs (36) and (38b) can be neglected. In this case, we can set $m^v = 0$ for $v \geq 2$ and it is straightforward to calculate $F_1(m^1)$. The result is Eq (38) with $\alpha R = 0$:

$$F_1(m^1) = \frac{1}{\gamma(1-\gamma)} \sum_{k=0,1} P_k(k-\gamma)\hat{\phi}((k-\gamma)m + Ik). \tag{40}$$

Note that the mean-field dynamics Eqs (39) and (40) in the small-load limit ($\alpha = P/N \to 0$ as $N \to \infty$) is exact.

For large network load $\alpha$ (i.e. $\alpha = O(1)$ as $N \to \infty$), the effect of background patterns may not be negligible. As shown above, the equilibrium solution in this case is given by Eqs (37) and (38). For the non-stationary dynamics Eq (39), we still use Eq (38) for $F_1(m^1)$ even though this equation has been derived under the assumption of stationarity. This means that we assume that the overlaps with background patters are always at their equilibrium value while the overlap variables with retrieved patterns evolve in time. While this assumption is not strictly true, it gives results in excellent agreement with full network simulations (Fig 2C). In other words, the mean-field dynamics in Fig 2C is correct before stimulus onset and after the system has retrieved pattern 1, whereas during transients the dynamics with $F_1(m^1)$ given by Eq (38) is an approximation. Moreover, we argued in the discussion that assuming a small or even negligible network load $\alpha \geq 0$ is a biologically plausible assumption for the human MTL. In this case, the dynamical mean-field equations for $\alpha = 0$, Eqs (39) and (40), are valid.

## Mean-field equations for two overlapping patterns

Overlap between two engrams is implemented as two patterns with a non-zero Pearson correlation coefficient. Without loss of generality, we take patterns $\vec{\xi}^1$ and $\vec{\xi}^2$ to be correlated, while all other $P - 2$ patterns are independent. We define the correlation $C$ between the two patterns as the Pearson correlation coefficient (covariance/variance):

$$C = \frac{\text{Cov}(\xi_i^1, \xi_i^2)}{\text{Var}(\xi_i^\mu)} = \frac{P_{11} - \gamma^2}{\gamma(1-\gamma)}, \tag{41}$$

where $P_{11} = P(\xi_i^1 = 1, \xi_i^2 = 1) = \langle \xi_i^1 \cdot \xi_i^2 \rangle$ is the joint probability of a neuron to be selective to both patterns. We generate correlated patterns with mean activity $\langle \xi_i^1 \rangle_i = \langle \xi_i^2 \rangle_i = \gamma$ and correlation coefficient $C$, using the procedure described in SI. The fraction $c$ of shared neurons is related to $C$ by the identity $c = C(1-\gamma) + \gamma$.

We are interested in the retrieval dynamics of the correlated patterns $\vec{\xi}^1$ and $\vec{\xi}^2$.

The derivation of the system of mean-field equations in case two correlated pattern case in Eq (43) is analogous to that described in the section above. To that end, we also assume that the stimulus only depends on the selectivities $\xi_i^1$ and $\xi_i^2$ of the neuron, i.e. $\hat{I}_i(t) = I_{\xi_i^1, \xi_i^2}(t)$ for all neurons $i = 1, \ldots, N$. The input term $h_i$ has now two non-negligible terms, both from

$\vec{\xi}^1$ and $\vec{\xi}^2$:

$$\hat{h}_i(m^1, m^2) = (\xi_i^1 - \gamma)m^1 + (\xi_i^2 - \gamma)m^2 + \sqrt{\alpha R(m^1, m^2)}Z_i + I_{\xi_i^1, \varsigma_i^2}(t), \tag{42}$$

where

$$\hat{\tau}\frac{dm^1}{dt} = -m^1 + \frac{1}{\gamma(1-\gamma)}\langle(\xi_i^1 - \gamma)\hat{\phi}(\hat{h}_i)\rangle_i \tag{43a}$$

$$\hat{\tau}\frac{dm^2}{dt} = -m^2 + \frac{1}{\gamma(1-\gamma)}\langle(\xi_i^2 - \gamma)\hat{\phi}(\hat{h}_i)\rangle_i \tag{43b}$$

$$q = \langle\hat{\phi}'(\hat{h}_i)\rangle_i \tag{43c}$$

$$p = \langle\hat{\phi}^2(\hat{h}_i)\rangle_i \tag{43d}$$

$$R(m^1, m^2) = \frac{p}{(1-q)^2}, \tag{43e}$$

and $Z_i \sim N(0, 1)$, $i = 1, \ldots, N$ are independent, standard normal random variables. Analogous to Eq (38) we compute the population averages in Eq (43) explicitly leading to the mean-field dynamics

$$\hat{\tau}\frac{dm^1}{dt} = -m^1 + F_1(m^1, m^2) \tag{44a}$$

$$\hat{\tau}\frac{dm^2}{dt} = -m^2 + F_2(m^1, m^2). \tag{44b}$$

Here, the nonlinear functions $F_1$ and $F_2$ are given by ($\mu = 1, 2$)

$$F_\mu(m^1, m^2) = \sum_{x^1=0,1}\sum_{x^2=0,1}\frac{x^\mu - \gamma}{\gamma(1-\gamma)}P_{x^1 x^2}\int\frac{dz}{\sqrt{2\pi}}e^{-\frac{z^2}{2}}\hat{\phi}(\hat{h}_{x^1 x^2}(m^1, m^2, z)) \tag{44c}$$

with

$$\hat{h}_{x^1 x^2}(m^1, m^2, z) = \sum_{\nu=1,2}(x^\nu - \gamma)m^\nu + I_{x^1, x^2}(t) + \sqrt{\alpha R_h(m^1, m^2)}z. \tag{44d}$$

This function can be interpreted as the mean-field input potential of a neuron with selectivity $\xi_i^1 = x^1$ and $\xi_i^2 = x^2$, background variability $Z_i = z$, in the case when the network has overlap $m^1$ and $m^2$ with patterns 1 and 2, respectively. The last term in Eq (44d) captures the influence of background patterns on the mean-field dynamics of $m^1(t)$ and $m^2(t)$. This influence is quantified by the function $R_h(m^1, m^2)$ representing the mean squared overlap of the system with the background patterns $\mu = 3, \ldots, P$. We used a subscript $h$ for $R_h(m^1, m^2)$ to indicate that $R$ depends functionally on the mean-field potential $\hat{h}_{x^1 x^2}(m^1, m^2, z)$. This functional is given by

$$R_h(m^1, m^2) = \frac{p}{(1-q)^2} \tag{44e}$$

$$q = \sum_{x^1=0,1}\sum_{x^2=0,1}P_{x^1, x^2}\int\hat{\phi}'(\hat{h}_{x^1 x^2}(m^1, m^2, z))e^{-\frac{z^2}{2}}\frac{dz}{\sqrt{2\pi}} \tag{44f}$$

$$p = \sum_{x^1=0,1}\sum_{x^2=0,1}P_{x^1, x^2}\int\hat{\phi}^2(\hat{h}_{x^1 x^2}(m^1, m^2, z))e^{-\frac{z^2}{2}}\frac{dz}{\sqrt{2\pi}}. \tag{44g}$$

The mean-field input potentials $\hat{h}_{x^1,x^2}(m^1, m^2, z)$, $x^1, x^2 \in \{0, 1\}$, needed in Eq (44c) are obtained from the self-consistent solution of the functional Eqs (44d)–(44g), details are in Section "Numerical solutions". Eq (44) simplify significantly for $\alpha = 0$, which is the parameter choice of most figures, so it is worth writing explicitly the $m^1$ and $m^2$ dynamics in the case of negligible load:

$$
\begin{aligned}
\hat{\tau}\frac{dm^1}{dt} = \quad &-m^1 + \frac{1}{\gamma(1-\gamma)}\Big\{P_{11}(1-\gamma)\hat{\phi}[(1-\gamma)(m^1+m^2)+I_1+I_2]+ \\
&P_{10}(1-\gamma)\hat{\phi}[(1-\gamma)m^1 - \gamma m^2 + I_1]- \\
&P_{01}\gamma\hat{\phi}[-\gamma m^1 + (1-\gamma)m^2 + I_2] - P_{00}\gamma\hat{\phi}[-\gamma(m^1+m^2)]\Big\},
\end{aligned}
\tag{45a}
$$

$$
\begin{aligned}
\hat{\tau}\frac{dm^2}{dt} = \quad &-m^2 + \frac{1}{\gamma(1-\gamma)}\Big\{P_{11}(1-\gamma)\hat{\phi}[(1-\gamma)(m^1+m^2)+I_1+I_2]- \\
&P_{10}\gamma\hat{\phi}[-\gamma m^1 + (1-\gamma)m^2 + I_1]+ \\
&P_{01}(1-\gamma)\hat{\phi}[(1-\gamma)m^1 - \gamma m^2 + I_2] - P_{00}\gamma\hat{\phi}[-\gamma(m^1+m^2)]\Big\}.
\end{aligned}
\tag{45b}
$$

Here, we have used the specific form $I_{x^1, x^2}(t) = I_1(t)x^1 + I_2(t)x^2$, $x^1, x^2 \in \{0, 1\}$, of the external currents, where the coefficients $I_1(t)$ and $I_2(t)$ are the external input currents given selectively to the neurons of pattern 1 and 2, respectively.

The same procedure can be generalized to generate several correlated binary patterns, as in Fig 2B. The generalization is straightforward, we can re-write the system in Eq (44) with one dynamical equations for each correlated pattern and add the relative terms in the input $\hat{h}(x^1, .., x^\mu, z)$. Finally we need the joint probabilities $P_{x^1,x^2,x^3}$ and $P_{x^1,x^2,x^3,x^4}$. The general formula for the joint probability is given in Eq (97) below. For instance, for three correlated patterns, the mean-field dynamics analogue to Eq (44) is given by

$$
\hat{\tau}\frac{dm^1}{dt} = -m^1 + \frac{1}{\gamma(1-\gamma)}\langle(\xi_i^1 - \gamma)\hat{\phi}(\hat{h}_i)\rangle_i
\tag{46a}
$$

$$
\hat{\tau}\frac{dm^2}{dt} = -m^2 + \frac{1}{\gamma(1-\gamma)}\langle(\xi_i^2 - \gamma)\hat{\phi}(\hat{h}_i)\rangle_i
\tag{46b}
$$

$$
\hat{\tau}\frac{dm^3}{dt} = -m^3 + \frac{1}{\gamma(1-\gamma)}\langle(\xi_i^2 - \gamma)\hat{\phi}(\hat{h}_i)\rangle_i
\tag{46c}
$$

$$
q = \langle\hat{\phi}'(\hat{h}_i)\rangle_i
\tag{46d}
$$

$$
p = \langle\hat{\phi}^2(\hat{h}_i)\rangle_i
\tag{46e}
$$

$$
R(m^1, m^2, m^3) = \frac{p}{(1-q)^2}
\tag{46f}
$$

where

$$
\hat{h}_i(m^1, m^2, m^3) = (\xi_i^1 - \gamma)m^1 + (\xi_i^2 - \gamma)m^2 + (\xi_i^3 - \gamma)m^3 + \sqrt{\alpha R(m^1, m^2, m^3)}Z_i + I_i.
\tag{47}
$$

## Excluding self-interaction

In Section "Review: mean-field equations for independent patterns" we show the derivation of the mean-field equations for the retrieval of one pattern in an attractor neural network with self-connections ("autapses"). To make the network more biologically plausible and to avoid the creation of local minima around the attractors corresponding to the stored patterns, we now consider the case where self-interactions are excluded [38]. The effect of excluding the self-interaction term on input terms in Eqs (38a) and (44d) is captured by a the correction term [38]:

$$\frac{q\alpha\hat{\phi}(\hat{h})}{(1-q)}. \tag{48}$$

Then, Eq (44d) becomes

$$\hat{h}_{x^1 x^2}(m^1, m^2, z) = (x^1 - \gamma)m^1 + (x^2 - \gamma)m^2 + \frac{q\alpha\phi(\hat{h}_{x^1 x^2}(m^1, m^2, z))}{(1-q)} + \sqrt{\alpha r}z + I_{x^1, x^2}(t), \tag{49}$$

where again $I_{x^1, x^2}(t) = I_1(t)x^1 + I_2(t)x^2$ and $x^1, x^2 \in \{0, 1\}$. In our simulation, we used the same stimulation for both patterns, i.e. $I_1(t) = I_2(t) \equiv I(t)$. The input term in Eq (49), is solved recursively in Fig 2B, left hand side.

## Overlapping background patterns

In Fig 2B we explore the possibility that the maximal fraction of shared neurons $c_{\max}$ might be influenced by the presence of Pearson's correlation between pairs of background patterns. Moreover, the assumption that there are many subgroups of overlapping memory engrams seems more biologically plausible. If we let the background patterns to be overlapping in sub-groups of 2 patterns each, the variable $R$ in the mean-field equations of Eq (43) needs to be replaced by In this section, we provide the derivation of the critical correlation in the presence of correlations between background patterns (main text, Fig 2B left).

To start with, let us suppose that each pattern is correlated with just one other, so a given pattern $\vec{\xi}^\nu$ is only correlated with one other pattern $\vec{\xi}^{\nu'}$, $\nu \neq \nu'$. In the following, the prime notation $\nu'$ denotes for any given pattern $\nu$ the index of the associated correlated pattern. What changes, compared to the derivation in Section "Review of attractor theory", is the variance of the heterogeneity term in Eq (25), $\langle\sigma^2\rangle = \sum_{\mu,\nu>2}\langle(\xi_i^\mu\xi_i^\nu - \gamma\xi_i^\nu - \gamma\xi_i^\mu + \gamma^2)\rangle m^\mu m^\nu$, where patterns are pair-wise correlated. For a fixed pair $(\nu, \nu')$, we obtain

$$\langle(\xi_i^\mu\xi_i^\nu - \gamma\xi_i^\nu - \gamma\xi_i^\mu + \gamma^2)\rangle = \delta_{\mu\nu}\gamma(1-\gamma) + \delta_{\mu,\nu'}(P_{11} - \gamma^2), \tag{50}$$

$$= \gamma(1-\gamma)[\delta_{\mu,\nu} + \delta_{\mu,\nu'}C], \tag{51}$$

where the second term at the right hand side captures the effect of correlation. Background patterns can still be approximated by a Gaussian variable in the large network limit, in this case with variance:

$$\langle\sigma^2\rangle = \gamma(1-\gamma)\sum_{\nu=3}^{P}[(m^\nu)^2 + Cm^\nu m^{\nu'}]. \tag{52}$$

In order to compute Eq (52), we need to derive $(m^\nu)^2$ and $m^\nu m^{\nu'}$. In what follows, we use the same definition of $q$ and $p$ as in Eq (38). Let us start from writing $m^\nu$ at the first-order

Taylor expansion for $m^\nu$ and $m^{\nu'}$ both small:

$$F_\nu(m^1, \ldots, m^P) \approx \frac{1}{\gamma(1-\gamma)N} \sum_{i=1}^N (\xi_i^\nu - \gamma)\hat{\phi}(\hat{h}_i) + \frac{1}{\gamma(1-\gamma)N} \sum_{i=1}^N (\xi_i^\nu - \gamma)^2 \hat{\phi}'(\hat{h}_i)m^\nu$$
$$+ \frac{1}{\gamma(1-\gamma)N} \sum_{i=1}^N (\xi_i^\nu - \gamma)\left(\xi_i^{\nu'} - \gamma\right)\hat{\phi}'(\hat{h}_i)m^{\nu'} \tag{53}$$

Then, following analogous passages as Eqs (26–29) we obtain the expressions:

$$(1-q)m^\nu = \frac{1}{\gamma(1-\gamma)N} \sum_{i=1}^N (\xi_i^\nu - \gamma)\hat{\phi}(\hat{h}_i) + qCm^{\nu'}, \tag{54a}$$

$$(1-q)m^{\nu'} = \frac{1}{\gamma(1-\gamma)N} \sum_{i=1}^N \left(\xi_i^{\nu'} - \gamma\right)\hat{\phi}(\hat{h}_i) + qCm^\nu. \tag{54b}$$

Eq (54) is a linear system of the form:

$$Dm^\nu = B + qCm^{\nu'} \tag{55a}$$

$$Dm^{\nu'} = B' + qCm^\nu \tag{55b}$$

where $B = \frac{1}{\gamma(1-\gamma)N} \sum_{i=1}^N \left(\xi_i^\nu - \gamma\right)\hat{\phi}(\hat{h}_i)$, $D = (1-q)$, similarly, $B' = \frac{1}{\gamma(1-\gamma)N} \sum_{i=1}^N \left(\xi_i^{\nu'} - \gamma\right)\hat{\phi}(\hat{h}_i)$, and $C = \frac{P_{11}-\gamma^2}{\gamma(1-\gamma)}$. System Eq (54) has solutions

$$m^\nu = \frac{DB + qCB'}{D^2 - C^2} \tag{56a}$$

$$m^{\nu'} = \frac{DB' + qCB}{D^2 - C^2}. \tag{56b}$$

We are now ready to write the expressions for $(m^\nu)^2$ and $m^\nu m^{\nu'}$:

$$(m^\nu)^2 = \frac{D^2 B^2 + (qC)^2 (B')^2 + 2DqCBB'}{(D^2 - (qC)^2)^2},$$
$$m^\nu m^{\nu'} = \frac{D^2 BB' + qCD(B')^2 + qCDB^2 + C^2 BB'}{(D^2 - (qC)^2)^2}, \tag{57}$$

where $B$ and $B'$ are analogous to the term on right hand side of Eq (27): $(B')^2 = B^2 = \frac{P}{N\gamma(1-\gamma)}$. Note that $B^2$ and $B'^2$ are equal in expectation, however $BB' \neq B^2$ in expectation due to the correlation between $\xi^\nu$ and $\xi^{nu'}$. The last missing piece is the cross term $BB'$ which can also be

calculated analogously to Eq (32):

$$
\begin{aligned}
BB' &= \frac{1}{[N\gamma(1-\gamma)]^2} \sum_{i=1}^{N}\sum_{j=1}^{N}\left[\left(\xi_i^v - \gamma\right)\left(\xi_j^{v'} - \gamma\right)\hat{\phi}(\hat{h}_i)\hat{\phi}(\hat{h}_j)\right] = \\
&= \frac{1}{[N\gamma(1-\gamma)]^2} \sum_{i=1}^{N}[(P_{11} - \gamma^2)]\hat{\phi}^2(\hat{h}_i) = \\
&= \frac{P_{11} - \gamma^2}{N[\gamma(1-\gamma)]^2}p = \\
&= \frac{Cp}{N\gamma(1-\gamma)}.
\end{aligned}
\tag{58}
$$

In the first passage, we used the fact that the first order approximations of $\hat{\phi}(\hat{h}_i)$ and $\hat{\phi}(\hat{h}_j)$ are independent (see Eq (53). Plugging the expressions for $(m^v)^2$ and $m^v m^{v'}$ into Eq (52), we obtain the variance

$$
\begin{aligned}
\langle\sigma^2\rangle &= \frac{\gamma(1-\gamma)}{(D^2 - q^2C^2)^2}\cdot \\
&\cdot\sum_{v\geq 3}\{D^2B^2 + q^2C^2(B')^2 + 2BDqCB' + CDBB' + DqC^2B^2 + DqC^2(B')^2 + q^2C^2Bb'\} = \\
&= \frac{\alpha p}{(D^2 - q^2C^2)^2}[D^2 + q^2C^2 + 4DqC^2 + q^2C^4]
\end{aligned}
\tag{59}
$$

Finally we can write the expression for the effective $R = \langle\sigma^2\rangle/\alpha$, under the effect of pairwise correlation between background patterns:

$$
\begin{aligned}
R &= \frac{p}{[D^2 - (qC)]^2}\gamma(1-\gamma)[D^2 + q^2C^2 + 4DqC^2 + q^2C^4] = \\
&= \frac{p}{[D^2 - (qC)]^2}\left[(1-q)^2 + q^2C^2 + 4(1-q)qC^2 + (1-q)^2C^4\right].
\end{aligned}
\tag{60}
$$

The so obtained expression for $R$ can be substituted to that of system Eq (43).

The derivation of Eq (60) can be extended to the case in which background patterns share correlation $C$ between non overlapping groups of exactly $p$ patterns. To do so, we need to extend the system in Eq (55b) linearly:

$$
M = \begin{pmatrix}
D & -qC & -qC & \cdots & -qC \\
-qC & D & -qC & \cdots & -qC \\
-qC & -qC & D & \cdots & -qC \\
\vdots & \vdots & \vdots & \ddots & \vdots \\
-qC & -qC & -qC & \cdots & D
\end{pmatrix}\cdot\begin{pmatrix}
m^v \\
m^{v'} \\
m_{v''} \\
\vdots \\
m_{v^{n'}}
\end{pmatrix} = \begin{pmatrix}
B \\
B' \\
B'' \\
\vdots \\
B^{n'}
\end{pmatrix}
\tag{61}
$$

where $M$ is a $p \times p$ matrix. In order to find the solution $\vec{m}^v = M^{-1}\vec{B}$ of system Eq (61) we need

to invert the matrix $M$. Indeed matrices $M$ of the form

$$\{M\}_{ij} = \begin{cases} D & \text{if } i = j \\ -qC & \text{if } i \neq j \end{cases} \tag{62}$$

are invertible. In order to derive the inverted matrix we can rewrite the matrix $M$ as $M = A - qCvv^T$, where $A$ is diagonal with entries $A_{i,i} = D - qC$ and $v$ is a column vector of all ones. If $M$ and $A$ are both invertible, we can use the Sherman-Morrison formula:

$$M^{-1} = (A - qCvv^T)^{-1} = A^{-1} - \frac{-qCA^{-1}vv^TA^{-1}}{1 - qCv^TA^{-1}v}. \tag{63}$$

Since $A$ is diagonal, then $(A^{-1})_{i,i} = (A_{i,i})^{-1} = \frac{1}{D+qC}$. Then

$$\{M^{-1}\}_{ij} = \begin{cases} \dfrac{1}{D + qC} - \dfrac{1}{c(D - qC)^2} & \text{if } i = j \\ -\dfrac{1}{c(D - qC)^2} & \text{if } i \neq j \end{cases} \tag{64}$$

where the constant $c = -\frac{1}{qC} + n\frac{1}{D+qC}$. Terms can be re-arranged to obtain:

$$\{M^{-1}\}_{ij} = \frac{1}{Z}\begin{cases} D + qC - 1 & \text{if } i = j \\ -1 & \text{if } i \neq j \end{cases} \tag{65}$$

where

$$Z = \frac{-D + (n - 1)qC}{qC(D + qC)^2}. \tag{66}$$

As a final note, we consider the case in which all patterns are equally correlated, then

$$\langle \sigma^2 \rangle = \sum_{v \geq 3}\sum_{\mu \geq 3}\left[\frac{P}{N}\gamma(1 - \gamma)(m^v)^2 + \frac{P^2}{N}(P_{11} - \gamma^2)m^v m^\mu\right] \tag{67}$$

The second term in the variance diverges as $N \to \infty$ because $P = \alpha N$ unless $P_{11} = \gamma^2$. We conclude that, in the limit $N \to \infty$ and assuming that the ration between patterns and neurons is a finite constant $\alpha > 0$, it is not possible to allow a correlation $C > 0$ between all stored patterns.

## Extract numerically the maximal correlation

In order to numerically compute the maximal correlation, we use the bifurcation diagram in Fig 9B: the fixed points in the phase-plane are projected on the $m^1$-axis and their positions are plotted as $C$ increases. From the bifurcation diagram we can extract the value $C_{\max}$ at which the single retrieval states merge with the saddle points and disappear. Thus, at $C = C_{\max}$ we have a saddle-node bifurcation (see the derivation of Eq (4)).

The value of the maximal correlation $C_{\max}$ can be calculated analytically in the limit of infinite steepness $b \to \infty$, vanishing load $\alpha = 0$, vanishing sparseness and load, $\gamma \to 0$ and $\alpha = 0$. This value matches the one extracted from the bifurcation plot in Fig 9C.

## Mean-field dynamics in the presence of adaptation and global feedback

In order to derive the mean-field equations for the model with adaptation and global feedback, we consider the simplest case, in which only two patterns are correlated ($\vec{\xi}^1$ and $\vec{\xi}^2$) while all the others are independent. Analogously to Section "Mean field equations for two correlated

patterns", we can group neurons into four homogeneous populations (in the presence of background patterns, the neural populations will be slightly inhomogeneous): neurons that are selective to both patterns ($\xi_i^1 = \xi_i^2 = 1$), neurons selective to pattern 1 but not 2 ($\xi_i^1 = 1, \xi_i^2 = 0$), neurons selective to pattern 2 but not 1 ($\xi_i^1 = 0, \xi_i^2 = 1$) and neurons that are selective to neither pattern 1 or 2 ($\xi_i^1 = \xi_i^2 = 0$). The probability for a neuron to belong to population $(x^1, x^2)$, i.e. $\xi_i^1 = x^1$ and $\xi_i^2 = x^2$, is the joint probability $P_{x^1 x^2}$ in Eq (93). Furthermore each population $(x^1, x^2)$ is characterized by a different firing threshold $\theta_{x^1 x^2}(t)$. Analogous to the derivation of Eq (44), we obtain the six-dimensional mean-field dynamics:

$$\hat{\tau} \frac{dm^1}{dt} = -m^1 + F_1(m^1, m^2, \{\hat{\theta}_{x^1 x^2}\}), \tag{68a}$$

$$\hat{\tau} \frac{dm^2}{dt} = -m^2 + F_2(m^1, m^2, \{\hat{\theta}_{x^1 x^2}\}), \tag{68b}$$

$$\hat{\tau}_\theta \frac{d\hat{\theta}_{x^1 x^2}}{dt} = -\theta_{x^1 x^2} + \hat{\theta}_0 + \hat{D}_\theta \hat{r}_{x^1 x^2}(m^1, m^2, \hat{\theta}_{x^1 x^2}), \qquad x^1, x^2 \in \{0, 1\}. \tag{68c}$$

Here, we have introduced the nonlinear functions

$$F_\mu(m^1, m^2, \{\hat{\theta}_{x^1 x^2}\}) = \sum_{x^1=0,1} \sum_{x^2=0,1} P_{x^1, x^2} \frac{x^\mu - \gamma}{\gamma(1-\gamma)} \hat{r}_{x^1 x^2}(m^1, m^2, \hat{\theta}_{x^1 x^2}), \qquad \mu = 1, 2 \tag{68d}$$

$$\hat{r}_{x^1 x^2}(m^1, m^2, \hat{\theta}_{x^1 x^2}) = \int \hat{\phi}(\hat{h}_{x^1 x^2}(m^1, m^2, \hat{\theta}_{x^1 x^2}, z)) e^{-\frac{z^2}{2}} \frac{dz}{\sqrt{2\pi}}, \tag{68e}$$

with the mean-field input potential

$$\begin{aligned} \hat{h}_{x^1, x^2}(m^1, m^2, \{\hat{\theta}_{x^1 x^2}\}, z) &= (x^1 - \gamma)m^1 + (x^2 - \gamma)m^2 + \sqrt{\alpha R} z \\ &\quad - \hat{\theta}_{x^1 x^2} - \frac{\hat{J}_0(t)}{\gamma} \sum_{k_1=0,1} \sum_{k_2=0,1} P_{k_1, k_2} \hat{r}_{k_1 k_2}(m^1, m^2, \hat{\theta}_{k_1 k_2}). \end{aligned} \tag{68f}$$

and the mean squared overlap of background patterns $R$ given by

$$R = \frac{p}{(1-q)^2} \tag{68g}$$

$$q = \sum_{x^1=0,1} \sum_{x^2=0,1} P_{x^1, x^2} \int \hat{\phi}'\left(\hat{h}_{x^1, x^2}(m^1, m^2, \{\hat{\theta}_{x^1 x^2}\}, z)\right) e^{-\frac{z^2}{2}} \frac{dz}{\sqrt{2\pi}} \tag{68h}$$

$$p = \sum_{x^1=0,1} \sum_{x^2=0,1} P_{x^1, x^2} \int \hat{\phi}^2\left(\hat{h}_{x^1, x^2}(m^1, m^2, \{\hat{\theta}_{x^1 x^2}\}, z)\right) e^{-\frac{z^2}{2}} \frac{dz}{\sqrt{2\pi}} \tag{68i}$$

In order to obtain $\hat{r}_{x^1 x^2}(m^1, m^2, \hat{\theta}_{x^1 x^2})$ in Eqs (68c), (68d) and (68e)–(68i) need be solved self-consistently (for more details, see Section "Numerical Solutions").

Analogously to the previous section, we can extract numerically the minimal and maximal correlation using the bifurcation analysis described in the Supplementary Information (S2 Fig).

## Stability of the fixed points

In order to compute the stability of the fixed points in Fig 9, we compute the eigenvalues of the Jacobian matrix $J$ of the $m^1 - m^2$ dynamics at the point location in the $m^1 - m^2$ plane.

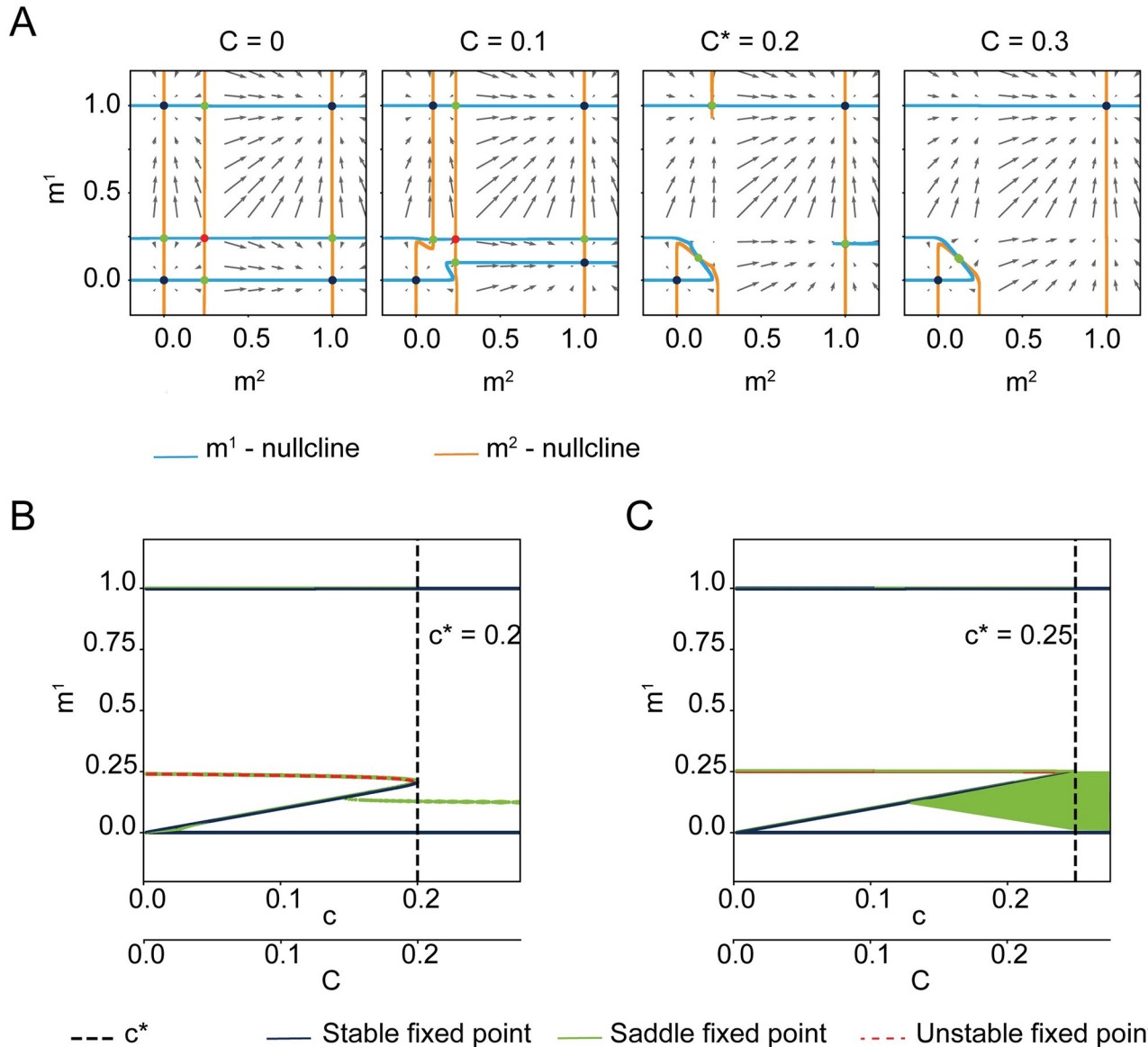

**Fig 9.** A) Four phase-planes of the dynamics of variables $m^1$ and $m^2$ for different values of correlation $C$. Fixed points are color-coded by their stability: blue = stable, green = saddle and red = unstable. B) Bifurcation diagram. The projection of the fixed points position on $m^1$ is plotted against $C$. The critical correlation $C_{\max}$ is highlighted by the black dashed line. C) Same as B, but in the limit $\hat{b} \to \infty$, which leads to $C \to \hat{h}_0$. Parameters: $\gamma = 0.002$, $\hat{b} = 100$, $\hat{h}_0 = 0.25$, $\alpha = 0$.

The Jacobian matrix is symmetric and the three independent entries are computed from Eq (43) as:

$$J_{11}(m^1, m^2) = \frac{\partial(-m^1 + F_1(m^1, m^2))}{\partial m^1}$$

$$= -1 + \frac{A}{\gamma(1-\gamma)} \langle (\xi_i^1 - \gamma)^2 \phi'(h_i) \rangle_i \tag{69}$$

$$J_{12}(m^1, m^2) \quad = \frac{\partial(-m^1 + F_1(m^1, m^2))}{\partial m^2}$$

$$= J_{21}(m^1, m^2) = \frac{A}{\gamma(1-\gamma)} \langle (\xi_i^1 - \gamma)(\xi_i^2 - \gamma)\phi'(h_i) \rangle_i \tag{70}$$

$$J_{22}(m^1, m^2) \quad = \frac{\partial(-m^2 + F_2(m^1, m^2))}{\partial m^2}$$

$$= -1 + \frac{A}{\gamma(1-\gamma)} \langle (\xi_i^2 - \gamma)^2 \phi'(h_i) \rangle_i \tag{71}$$

In the numerical computation of the J, we exploited the symmetries under exchange of $m^1$ and $m^2$: $J_{22}(m^1, m^2) = J_{11}(m^2, m^1)$ and $J_{21}(m^1, m^2) = J_{12}(m^2, m^1)$.

Analogously to the system in Eq (43), also the Jacobian matrix can be adapted to the case of 3 or 4 correlated pattern, using the joint probabilities in Eq (97) and the generic forms

$$J_{\mu,\mu}(m^\mu, m^\mu) = -1 + \frac{A}{\gamma(1-\gamma)} \langle (\xi_i^\mu - \gamma)^2 \phi'(h_i) \rangle_i, \tag{72}$$

$$J_{\mu,\nu}(m^\mu, m^\nu) = \frac{A}{\gamma(1-\gamma)} \langle (\xi_i^\mu - \gamma)(\xi_i^\nu - \gamma)\phi'(h_i) \rangle_i \tag{73}$$

## The limit case $b \to \infty$ (Heaviside transfer function)

In the limit $b \to \infty$, the transfer function converges to the Heaviside step function $\phi(h) = r_{\max} \Theta(h - h_0)$ which leads to some simplifications in the explicit writing of the mean-field system Eq (43). First of all, we can rewrite $\phi^2(h) = r_{\max}^2 \Theta(h - h_0)$ and $\phi'(h) = r_{\max} \delta(h - h_0)$, where $\delta(x)$ is the Dirac delta function. In the dimension-less notation, we would then write $\phi(h) = \Theta(h - \hat{h}_0)$, $\phi^2(h) = \Theta(h - \hat{h}_0)$ and $\phi'(h) = \delta(h - \hat{h}_0)$, where $\delta(x)$ We can re-write Eq (43) as as follows:

$$\sum_{x^1=0,1x^2=0,1} P_{x^1x^2} \int \frac{dz}{\sqrt{2\pi}} e^{-\frac{z^2}{2}} \Theta(\hat{h}_{x^1x^2}(m^1, m^2, z) - \hat{h}_0) \tag{74}$$

$$\hat{\tau}\frac{dm^1}{dt} = -m^1 + \sum_{x^1=0,1x^2=0,1} \frac{x^1 - \gamma}{\gamma(1-\gamma)} P_{x^1x^2} \int \frac{dz}{\sqrt{2\pi}} e^{-\frac{z^2}{2}} \Theta(\hat{h}_{x^1x^2}(m^1, m^2, z) - \hat{h}_0) \tag{75a}$$

$$\hat{\tau}\frac{dm^2}{dt} = -m^2 + \sum_{x^1=0,1x^2=0,1} \frac{x^2 - \gamma}{\gamma(1-\gamma)} P_{x^1x^2} \int \frac{dz}{\sqrt{2\pi}} e^{-\frac{z^2}{2}} \Theta(\hat{h}_{x^1x^2}(m^1, m^2, z) - \hat{h}_0) \tag{75b}$$

$$q = \sum_{x^1=0,1x^2=0,1} P_{x^1x^2} \int \frac{dz}{\sqrt{2\pi}} e^{-\frac{z^2}{2}} \delta(\hat{h}_{x^1x^2}(m^1, m^2, z) - \hat{h}_0) \tag{75c}$$

$$p = \sum_{x^1=0,1x^2=0,1} P_{x^1x^2} \int \frac{dz}{\sqrt{2\pi}} e^{-\frac{z^2}{2}} \Theta(\hat{h}_{x^1x^2}(m^1, m^2, z) - \hat{h}_0) \tag{75d}$$

$$R(m^1, m^2) \quad = \frac{p}{(1-q)^2}, \tag{75e}$$

where

$$\hat{h}_i(m^1, m^2) = (\xi_i^1 - \gamma)m^1 + (\xi_i^2 - \gamma)m^2 + \sqrt{\alpha R(m^1, m^2)}Z_i + I_{\xi_i^1, \xi_i^2}(t).$$ (76)

In the next passage the erfc function come at hands. Erfc is defined as $\text{erfc}(x) = 1 - \text{erf}(x)$, where erf is the error function and we use the following identity, which follows directly from the definition:

$$\int_c^\infty \frac{e^{-\frac{x^2}{2}}}{\sqrt{2\pi}} dx = \frac{1}{2}\text{erfc}\left(\frac{c}{\sqrt{2}}\right).$$ (77)

The identity in Eq (77) allows to rewrite the system Eq (75) as:

$$\tau\frac{dm^1}{dt} = -m^1 + \frac{1}{2\gamma(1-\gamma)}\sum_{x^1}\sum_{x^2}P_{x^1,x^2}\left(\xi_i^1 - \gamma\right)\text{erfc}\left(\frac{h_0 - \hat{h}_i}{\sqrt{2\alpha R}}\right)$$ (78a)

$$\tau\frac{dm^2}{dt} = -m^2 + \frac{1}{2\gamma(1-\gamma)}\sum_{x^1}\sum_{x^2}P_{x^1,x^2}\left(\xi_i^2 - \gamma\right)\text{erfc}\left(\frac{h_0 - \hat{h}_i}{\sqrt{2\alpha R}}\right)$$ (78b)

$$q = \sum_{x^1}\sum_{x^2}P_{x^1,x^2}\frac{1}{\sqrt{2\pi}}e^{-\frac{(\hat{h}_i)^2}{\sqrt{2\alpha R}}}$$ (78c)

$$p = \frac{1}{2}\sum_{x^1}\sum_{x^2}P_{x^1,x^2}\text{erfc}\left(\frac{h_0 - \hat{h}_i}{\sqrt{2\alpha R}}\right)$$ (78d)

$$R = \frac{p}{(1-q)^2}.$$ (78e)

It is important to make a remark on the units of the system: if we do not use the unit-less notation, then the variable $q$ is proportional to $r_{max}$ and the variable $p$ is proportional to $r_{max}^2$.

If we consider the case where neural self-interaction is excluded, an extra correction term should be added to the input $h(x^1, x^2, z))$ and its limit for $b \to \infty$ reads as follows:

$$\frac{A^2 q\alpha\phi(h)}{(1 - Aq)} \underset{b\to\infty}{\to} \frac{A\alpha r_{max}}{2}.$$ (79)

In the dimensionless notation, the correction term is reduced to a constant $\frac{\alpha}{2}$ and we can write explicitly the input term $\hat{h}_{x^1 x^2}(m^1, m^2, z)$, when self interaction is excluded:

$$\hat{h}_{x^1 x^2}(m^1, m^2, z) = (x^1 - \gamma)m^1 + (x^2 - \gamma)m^2 + \frac{\alpha}{2} + \sqrt{\alpha r}z + I_{x^1, x^2}(t),$$ (80)

Finally, in order to derive the critical correlation let us consider the retrieving state of pattern 1 (that of pattern 2 is symmetric with respect to the $m^1 - m^2$ axis in absence of external input): in this state, $m^1 = 1$, and $m^2$ depends on the correlation $C$, as it emerges from Fig 9A, however what is the exact value? It can be computed analytically in the limit, $b \to \infty$ and $\gamma \to$

 

0. We rewrite the equation for $m^2$ in Eq 45 as:

$$\hat{\tau}\frac{dm^2}{dt} = \quad -m^2 + \left\{ \frac{1}{\gamma}P_{11}(1-\gamma)\hat{\phi}[(1-\gamma)(m^1+m^2)+I_1+I_2]- \right.$$

$$\frac{1}{1-\gamma}P_{10}\gamma\hat{\phi}[-\gamma m^1+(1-\gamma)m^2+I_1]+ \tag{81}$$

$$\left. \frac{1}{\gamma}P_{01}(1-\gamma)\hat{\phi}[(1-\gamma)m^1-\gamma m^2+I_2]-\frac{1}{1-\gamma}P_{00}\hat{\phi}[-\gamma(m^1+m^2)] \right\}.$$

Next we need to write the probabilities $P_{x^1,x^2}$ as a function of $\gamma$:

$$P_{11} = \gamma^2 + \gamma(1-\gamma)C \tag{82a}$$

$$P_{10} = P_{01} = P(x^2=0|x^1=1)P(x^1=1) = \gamma(1-\gamma) - C\gamma(1-\gamma) \tag{82b}$$

$$P_{00} = 1 - P_{11} - P_{10} - P_{01} - P_{00} = (1-\gamma)^2 + \gamma(1-\gamma)C. \tag{82c}$$

Then, in the limit $\gamma \to 0$, we have

$$\hat{\tau}\frac{dm^2}{dt} = -m^2 + \left\{ C\hat{\phi}[m^1+m^2+I_1+I_2] + (1-C)\hat{\phi}[m^2+I_2] - \hat{\phi}[0] \right\}. \tag{83}$$

Using the limit $b \to \infty$, in the assumption that we are recalling the first concept, $m^1 = 1$, there is not any external input, we obtain

$$m^2 = C\Theta(1+m^2-\hat{h}_0) + (1-C)\Theta(m^2-\hat{h}_0) - \Theta(-\hat{h}_0). \tag{84}$$

Since $\hat{h}_0 < 1$ and $m^2 \geq 0$, the term $\Theta(1+m^2-\hat{h}_0) = 1$. On the other hand, $\Theta(-\hat{h}_0) = 0$. Therefore, $m^2 = C$ if $m^2 < \hat{h}_0$ (cf. the bifurcation diagram in Fig 9C). In the limit case where $m^2 \to \hat{h}_0$ we obtain Eq (4):

$$C_{\max} \leq C_{\max} \equiv \hat{h}_0 = \frac{h_0}{Ar_{\max}}. \tag{85}$$

## How does a network embed groups of overlapping memories? Different algorithms to generate correlated patterns

In this section we describe how a single subgroup of $K$ patterns with sparseness $\gamma$ is created according to three different algorithms. Patterns belonging to the same subgroup correspond to associated concepts and share pair-wise a fraction of neurons $c$. For the hierarchical generative model and the indicator neuron model, we associate the algorithm to the theoretical probability distribution for a neuron to respond exactly to $k$ concepts out of $K$.

**Hierarchical generative model.** We start by creating a "parent" pattern which is not part of the subgroup. The parent pattern has sparseness $\lambda = \gamma/c$: prob $(\xi_i^{\text{parent}} = 1) = \lambda$. We proceed to create the actual patterns by copying the ones of the parent pattern with probability $c$, while the zeros stay untouched, following the conditional probabilities

$$\text{prob}(\xi_i^\mu = 1|\xi_i^{\text{parent}} = 1) = c, \tag{86a}$$

$$\text{prob}(\xi_i^\mu = 1|\xi_i^{\text{parent}} = 0) = 1 - c, \tag{86b}$$

 

$$\text{prob}(\xi_i^\mu = 1 | \xi_i^{\text{parent}} = 0) = 0, \tag{86c}$$

$$\text{prob}(\xi_i^\mu = 0 | \xi_i^{\text{parent}} = 0) = 1. \tag{86d}$$

This ensures that the patterns $\xi_i^\mu$ have the right sparseness and fraction of pair-wise shared neurons. The sparseness can be checked as follows:

$$\text{prob}(\xi_i^\mu = 1) = \lambda c = \gamma, \tag{87a}$$

$$\text{prob}(\xi_i^\mu = 0) = \lambda(1 - c) + (1 - \lambda) = 1 - \gamma. \tag{87b}$$

On the other hand, the fraction of pair-wise shared neurons is given by the conditional probability that a neuron is part of pattern $v$ given that is it part of pattern $\mu$:

$$\text{prob}(\xi_i^v = 1 | \xi_i^\mu = 1) = c + (1 - c)\delta^{\mu v}. \tag{88}$$

Hence the fraction of shared neurons as it should be. More generically, the theoretical probability (or the expectation) that a neuron participates in $k$ patterns out of $K$ is

$$P^K(k) = \frac{K!}{(K - k)!k!}\lambda c^k(1 - c)^{K - k} + (1 - \lambda)\delta_{k0}. \tag{89}$$

**Indicator neuron model.** To create a subgroup of pair-wise associated patterns using indicator neurons (i.e. neurons that indicate the subgroup), we proceed in three steps:

1. generate with probability $\lambda$ a small subset of indicator neurons for this subgroup. This subset gives a parent pattern of indicator neurons:

$$\text{prob}(\xi_i^{\text{parent}} = 1) = \lambda_{\text{ind}} = \frac{c\gamma - \gamma^2}{(1 - \epsilon)^2 - 2\gamma(1 - \epsilon) + c\gamma}. \tag{90}$$

In a network of $N$ neurons, $n_{\text{ind}} = \lambda_{\text{ind}} N$ are indicator neurons.

2. To create each pattern $\mu$ of the subgroup, copy indicator neurons with probability $(1 - \epsilon)$:

$$\text{prob}(\xi_i^\mu = +1 | \xi_i^{\text{parent}} = 1) = 1 - \epsilon \tag{91}$$

3. Add random neurons (with probability $\Omega$) to pattern $\mu$

$$\text{prob}(\xi_i^\mu = 1 | \xi_i^{\text{parent}} = 0) = \Omega = \frac{\gamma - \lambda_{\text{ind}}(1 - \epsilon)}{1 - \lambda_{\text{ind}}}. \tag{92}$$

This last probability can also be interpreted as the probability of flipping a 0 from the parent pattern when creating the correlated patterns.

With this construction, the total number of neurons that are active in pattern $\mu$ is $\lambda_{\text{ind}}N(1 - \epsilon) + (1 - \lambda_{\text{ind}})N\frac{\gamma - \lambda_{\text{ind}}(1 - \epsilon)}{1 - \lambda_{\text{ind}}} = N\gamma$ as it should be. The value of $\lambda_{\text{ind}}$ is chosen in order to ensure that the fraction of pair-wise shared neurons is $c$. Indeed we found it by solving $c\gamma = \lambda(1 - \epsilon)^2 + (1 - \lambda)\Omega$.

In this work, we always choose $\epsilon$ such that $\epsilon = \Omega$, For specific case $\epsilon = \Omega$, it is possible to derive $\epsilon$ directly from the correlation $C$ and the sparsity $\gamma$.

We create a "parent" pattern $\vec{\xi}^0$ with mean activity $\langle \xi_i^0 \rangle_i = \lambda$. Starting from $\vec{\xi}^0$ we create $\vec{\xi}^1$ and $\vec{\xi}^2$, each unit $i$ has probability $\epsilon$ of being the equal to $\xi_i^0$ and probability $1 - \epsilon$ of being flipped compared to $\xi_i^0$. All other patterns $\xi^\mu, \mu = 3, \ldots, P$ are sorted independently from a Bernoulli distribution with probability $P(\xi_i^\mu = 1) = \gamma$. The joint probabilities $P_{kl} = P(\xi_i^1 = k, \xi_i^2 = l)$ can be computed as functions of the probabilities $\lambda$ and $\epsilon$:

$$P_{11} = \lambda \epsilon^2 + (1 - \lambda)(1 - \epsilon)^2, \tag{93a}$$

$$P_{10} = P_{01} = \lambda \epsilon (1 - \epsilon) + (1 - \lambda) \epsilon (1 - \epsilon) = \epsilon (1 - \epsilon), \tag{93b}$$

$$P_{00} = \lambda (1 - \epsilon)^2 + (1 - \lambda) \epsilon^2. \tag{93c}$$

Note that by this procedure we only obtain non-negative correlations $C \in [0, 1]$.
Using $P_{11}$ from Eq (93), we can express $C$ as

$$C(\lambda, \epsilon) = \frac{P_{11} - \gamma^2}{\gamma(1 - \gamma)} = \frac{(1 - \lambda)[\lambda \epsilon^2 + (1 - \epsilon)^2]}{\gamma(1 - \gamma)}. \tag{94}$$

Similarly, the mean activity of the correlated patterns can be expressed as a function of $\lambda$ and $\epsilon$ as

$$\gamma(\lambda, \epsilon) = \langle \xi_i^1 \rangle_i = \langle \xi_i^2 \rangle_i = \lambda \epsilon + (1 - \lambda)(1 - \epsilon). \tag{95}$$

So far, we showed how to generate correlated patterns given the probabilities $\lambda$ and $\epsilon$. Conversely, how do we choose $\lambda$ and $\epsilon$ given the mean activity $\gamma$ and the correlation $C$, $C \geq 0$? To this end, we invert the above relations in order to solve for $\lambda(C, \gamma)$ and $\epsilon(C, \gamma)$:

$$\lambda = \frac{\gamma + \epsilon - 1}{2\epsilon - 1}, \tag{96a}$$

$$2\epsilon^3 - 3\epsilon^2 + [1 + 2\gamma(1 - \gamma)(1 - \hat{C})]\epsilon - \gamma(1 - \gamma)(1 - \hat{C}) = 0, \tag{96b}$$

Eq (96b) has up to three solutions, we chose those that are real and in the range [0, 1].

The same procedure can be generalized to generate several correlate binary patterns. The general formula for the joint probability can be written as follows:

$$P_{x^1,\ldots,x^n} = \lambda \epsilon^a (1 - \epsilon)^b + (1 - \lambda) \epsilon^b (1 - \epsilon)^a, \tag{97}$$

where $a = \sum_{\mu=1}^n x^\mu$ is the number of $x^\mu$ variables taking value 1 and $b = n - a$ is the number of $x^\mu$ variables taking value 0. The value of the joint probabilities in Eq (97) is invariant under permutation of the $x^\mu$.

**Iterative overlap-generating model.** In this subgroup construction, we do not define any parent pattern. We define the number of active neurons as $\gamma N$ and the number of pair-wise shared neurons as $\gamma c N$.

1) We define the set of "untouched neurons", which counts all neurons at the beginning of the procedure

2) To create pattern 1 we randomly sample $\gamma N$ neurons and exclude the sampled neurons from the untouched ones.

We follow the iterative steps, from 3) to 5), to create patterns 2 to p.

3) For every pattern $\xi^\mu$ with $\mu$ from 2 to $K$, compare it with each of the already created patterns. Let's suppose we are comparing the new pattern $\mu$ with the already formed pattern $\nu$. a) check how many neurons are in common between the two. b) sample from pattern $\nu$ the remaining neurons needed to reach $\gamma c N$ shared neurons.

4) Complete pattern $\mu$ by adding neurons from the untouched ones until reaching $\gamma N$ active units.

5) Remove the units used in point 4 from the untouched ones.

It is important to underline the necessity of point 3a). To illustrate this point, let us consider the case we are building a subgroup of 3 patterns. We build the first one as in point 2. When we build pattern two starting from scratch, it does not share any neuron with pattern 1, so we just sample $\gamma c N$ from pattern 1 and $\gamma N(1 - c)$ from the untouched neurons. Now we move to pattern 3. As before, it does not share neurons from pattern 2, so we pick $\gamma c N$ from it. Now we compare pattern 3 with pattern 1: it can be that between the neurons we picked from pattern 2 some belong to pattern 1 as well, that's why we need to adjust the number of neurons to pick in order to preserve the correct amount of pair-wise correlation.

When the subgroup size $K$ is big however it is still possible to exceed the correct fraction of shared neurons between some of the patterns that are built last. Let's suppose we are creating a subgroup of size $K = 16$, I start by applying point 3 of the algorithm between pattern 16 and 15, then pattern 16 and 14 and so on. It can be that when we get to the point of picking neurons from pattern 4, 3, 2, 1 we take some neurons that also belong to pattern 15 but they are not the ones we picked in the previous iteration and thus get accepted. This creates a higher correlation between the last built patterns in large subgroups. We checked that this does not influence significantly the average pairwise correlation during the virtual experiments described in the next section.

## Comparing algorithm predictions with experimental data

The experimental dataset of Fig 6 comes from a previous publication [6]. Data were collected in 100 recording sessions with epileptic patients implanted with chronic depth electrodes in the MTL for the monitoring of epileptic seizures. Micro-wires recorded the localized neural activity; spike detection and sorting allowed to identify the activity of 4066 single neurons. During recordings, patients were shown different pictures of known people and places repeated several times. For each neuron, the stimuli eliciting a response were identified using a statistical criterion based on the modulations of firing rate during stimulus presentation compared to baseline epochs. For additional details on the dataset and data processing we refer to the original publication. The association between each pair of stimuli was estimated using a web-based association score.

In order to compare the predictions of the algorithms with the data, we try to reproduce the real data by running virtual experiments based on the three algorithms presented in the previous section. In each virtual experiment we replicate the conditions of the real experiment as follows. For each real experimental session, we first extract the number of responsive neurons in each session. We then group the presented stimuli into clusters based on an association matrix derived from the web-association scores. To do so, we use an hierarchical agglomerating clustering algorithm with threshold equal to the mean of the association matrix for the session. Such clusters indicate the amount and the size of the patterns subgroups we have to build for the corresponding virtual experiment.

We can then proceed with the virtual experiment: in each session we a) build subgroups of patterns in the same number and size as the clusters of stimuli for each of the three algorithms

and then b) sample a neuron at the time and count to how many patterns does it respond to. c) Finally, the count of how many stimuli a neuron responds to that of other sessions. We sample neurons until we match the number of responsive neurons with that of the real experimental session. Each virtual experiment counts $N = 10^5$ neurons and it is run 40 times and plot in Fig 6C the normalised mean and standard deviation.

We choose to ignore non-responding neurons in our analysis, since it is likely that the proportion of non-responsive neurons compared to that of responsive ones is largely underestimated in the experiment (non-responsive neurons are more likely to remain silent during the experiment and not to be recorded at all).

## Comparing virtual experiments and expected distributions

It is also possible to compare the virtual experiments with the theoretical distributions in Eqs (89) and (97). Eqs (89) and (97) provide the probability that a neuron is selective to $k$ out of $K$ patterns if a single subgroup of stimuli is stored in the network. But how do we combine such probabilities when several subgroups of patterns are stored in the network? We define $\Psi_s(k)$ the probability that a neuron responds to exactly $k$ patterns in session $s$. We know from the previous session the number and sizes $G_j$ of subgroups present in each session. Then

$$\Psi_s(k) = \sum_{j=k}^{\text{max}K} G_j P^j(k) \zeta_j \tag{98}$$

where max$K$ is the biggest between all subgroup sizes $K_j$ and $\zeta_j = 1 - P^j(0)$ is the probability that a neuron takes part into the subgroup $j$. The formula Eq 98 is valid in the assumption that subgroups are strictly disjoint, meaning that we assume that the same can not take part into encoding patterns belonging to different subgroups. This assumption is not true for the way we algorithmically build subgroups patterns in the virtual experiments, however dropping it make the expression for $\Psi_s(k)$ not treatable. Finally the probabilities $\Psi_s(k)$ from each session must be combined into the final distribution $\Psi_{\text{final}}(k)$:

$$\Psi_{\text{final}}(k) = \frac{\sum_s N_s^{\text{sample}} \Psi_s(k)}{\sum_s \sum_k N_s^{\text{sample}} \Psi_s(k)} = \frac{N_1^{\text{sample}} \Psi_1(k) + N_2^{\text{sample}} \Psi_2(k) + \ldots}{N_{\text{tot}}^{\text{sample}}} \tag{99}$$

where $N_s^{\text{sample}}$ is the amount of responsive neurons measured in each experimental session and $N_{\text{tot}}$ is the total amount of measured responsive neurons. In the last passage, note that $\sum_s \sum_k N_s^{\text{sample}} \Psi_s(k) = \sum_s N_s^{\text{sample}} \sum_k \Psi_s(k) = N_{\text{tot}}$, since $\Sigma_k \Psi_s(k)$ in every session $s$. The comparison between the theoretical distributions, the virtual experiments and the experimental data is shown in S6 Fig. The virtual experiments are the same a in Fig 6: we re-run the experiment 40 times and took the average (main points) and standard deviation (error bars). The small mismatch between the theoretical predictions and virtual experiments is due to the fact that in the theoretical prediction we do not allow the same neurons to take part to two or more subgroups of concepts, while there is no such a restriction in the virtual experiment. Theory prediction and mean of the virtual experiments are really close, proving that only very few neurons take part in encoding different subgroups.

## Heterogeneous frequency-current curves

The frequency-current function of model neurons is neuron-specific and re-written as

$$\phi_i(x) = \frac{(r_{\text{max}})_i - (r_{\text{min}})_i}{1 + e^{-\hat{b}(x - (\hat{h}_0)_i)}} + (r_{\text{min}})_i, \tag{100}$$

where the values of $(r_{\min})_i$ and $(r_{\max})_i$ are randomly sampled for each neuron from a Gaussian distribution with mean and standard deviation $\mu_{\min}$, $\sigma_{\min}$ and $\mu_{\max}$, $\sigma_{\max}$ respectively. The parameter $(\hat{h}_0)_i$ is then defined as $(\hat{h}_0)_i = h_0((r_{\max})_i - (r_{\min})_i)/(A\mu_{\max}^2)$, where $h_0$ is a global constant. Finally in the firing rate equation, we re-scale the firing rates as follows:

$$r_i \rightarrow \text{Max}\left[0, \frac{r_i - (r_{\min})_i}{(r_{\max})_i - (r_{\min})_i}\right]\mu_{\max}. \tag{101}$$

In Fig 7 we choose the parameters $\mu_{\min} = 0$ Hz, $\mu_{\max} = 1 = 40$ Hz, and $\sigma_{\min} = \sigma_{\max} = 4$ Hz.

## Diluted weight matrix

We define an attractor neural network of N units, where each unit receives input from K others. The probability of having a connection between two units is $d = M/N$. The load of the network is defined as $\alpha = P/N$, where P is the total number of patterns. We also assume that $A/d =$ constant (to be introduced into the dimensional analysis). The input term Eq 7 is filtered with transfer function $\phi$, which is chosen to be a sigmoid as in Eq 9. The connection matrix, $w_{ij}$, contains the synaptic weights between neurons $i$ and $j$, but, compared to Eq 5, connections are diluted with probability $d$ as defined in [24]

$$w_{ij} = \frac{A}{N\gamma(1 - \gamma)}\frac{d_{ij}}{d}\sum_{\mu}^{P}(\xi_i^\mu - \gamma)\left(\xi_j^\mu - \gamma\right) \tag{102}$$

where $d_{ij}$ is 1 with probability $M/N$ and 0 otherwise and the constant $A$ can be interpret as "connection strength". In order for the weights to have expectation $\langle w_{ij}\rangle = 0$, we subtract the mean activity of patterns $<\xi_i^\mu>= \gamma$. Using the similarity measure introduced in Eq 2, the input terms $h_j$ can also be re-written as a function of the overlaps $m^1, \ldots, m^P$, by using the definition of the weights $w_{ij}$, Eq (102), and that of the overlaps.

$$\begin{aligned}
h_i &= \sum_j^N w_{ij}r_j = \frac{A}{Nd\gamma(1 - \gamma)}\sum_j^N d_{ij}\sum_\mu^P(\xi_i^\mu - \gamma)\left(\xi_j^\mu - \gamma\right)r_j = \\
&= \frac{A}{Nd\gamma(1 - \gamma)}\sum_j^N d_{ij}(\xi_i^1 - \gamma)r_j + \frac{A}{Nd\gamma(1 - \gamma)}\sum_j^N d_{ij}\sum_{\mu=2}^P\left(\xi_j^\mu - \gamma\right)(\xi_i^\mu - \gamma)r_j
\end{aligned} \tag{103}$$

where we have separate the "signal" related to the first pattern being retrieved and a noise term $Y_i$. We write $h_i = Am^1 + Y_i$, with

$$Y_i = \frac{A}{Nd\gamma(1 - \gamma)}\sum_j^N(1 - d_{ij})(\xi_i^1 - \gamma)r_j + \frac{A}{Nd\gamma(1 - \gamma)}\sum_j^N d_{ij}\sum_{\mu=2}^P\left(\xi_j^\mu - \gamma\right)(\xi_i^\mu - \gamma)r_j \tag{104}$$

Since the terms $d_{ij}$ and $(\xi_i^1 - \gamma)$ are independent, we have $\langle Y_i\rangle$.

We assume $Y_i$ to be distributed like a Gaussian with variance

$$\langle\langle\sigma^2\rangle\rangle_i = \langle\langle(Y_i)^2\rangle\rangle_i$$

$$= \frac{1}{N}\sum_i^N \frac{A^2}{N^2 d^2 \gamma^2(1 - \gamma)^2}\sum_{\mu\neq 1}^P\sum_{v\neq 1}^P\left\langle(\xi_i^\mu - \gamma)(\xi_i^v - \gamma)\sum_j\sum_k\left(\xi_j^\mu - \gamma\right)(\xi_k^v - \gamma)\right\rangle d_{ij}d_{ik}r_j r_k \tag{105}$$

$$= \frac{1}{N}\sum_i^N \frac{A^2}{N^2 d^2 \gamma^2(1 - \gamma)^2}\sum_{\mu\neq 1}^P(\xi_i^\mu - \gamma)^2\sum_j d_{ij}(\xi_j^\mu - \gamma)^2 r_j^2$$

in the last passage, we used the fact that $\langle (\xi_j^\mu - \gamma)(\xi_k^\nu - \gamma) \rangle = \delta_{jk}\gamma(1 - \gamma)$ and $d_{ij}^2 = d_{ij}$. We then apply the same independence argument as used for the signal term and obtain

$$\langle \sigma^2 \rangle \quad = \frac{A^2 r_{\max}^2}{d^2} \gamma(1 - \gamma) d \sum_\mu (m^\mu)^2 \tag{106}$$

From now on the passages are the same as in the SI, except maybe the correction term for excluding self-interaction, which I should recompute.

The final difference in the equations is that the term $\sqrt{\alpha' r} z$, where $\alpha' = P/N$ should be substituted with $\sqrt{\alpha d r} z$. The two terms however are equivalent since $\alpha' = \alpha d$.

## Numerical solutions

The code used to generate the results of this work can be downloaded from: https://github.com/ChiaraGastaldi/pub_Gastaldi_2021_AttractorNetwork.git.

**Two correlated patterns: Finding the fixed points.** The system in Eq (43) is solved numerically to obtain the fixed nullclines, points, and flux arrows, plotted in Fig 9. Fixed points are obtain through a grid search in the three-dimensional space spanned by $m^1$, $m^2$ and $R$. For each value of $R_{\text{val}} \in [0, \max(R)]$ and $m_{\text{val}}^1, m_{\text{val}}^2 \in$ [Lower bound, Upper bound], Eqs (44d)–(44g) are solved. We call the value of $R$ obtained by Eq (44d) $R_{\text{reconstructed}}$. If $R_{\text{val}}$ and $R_{\text{reconstructed}}$ are close enough, namely

$$|R_{\text{val}} - R_{\text{reconstructed}}| < \text{ correction} - \text{constant} \cdot \text{step}. \tag{107}$$

The quantity called "step" is the step size of the linear space we used to span $R$,

$$\text{step} = \frac{\max(R)}{\text{Resolution}} \tag{108}$$

The correction constant can increase or decrease the range in which we accept a value $R_{\text{val}}$ as a valid solution: it is equal to 1 in most cases, but can be chose to be a bit bigger than one to avoid counting the same fixed point too many times. The values of $R_{\text{val}}$ that satisfy Eq (107) are then used to solve Eq (44), providing the values $m_{\text{reconstructed}}^1$ and $m_{\text{reconstructed}}^2$. Analogously to before, we find the solutions of Eq (44) comparing the values $m_{\text{val}}^1$ and $m_{\text{val}}^2$ with the recomputed counterparts $m_{\text{reconstructed}}^1$ and $m_{\text{reconstructed}}^2$ as follows

$$|m_{\text{val}}^\mu - m_{\text{reconstructed}}^\mu| < \text{ correction} - \text{constant} \cdot \text{step}, \tag{109}$$

where the step is defined as

$$\text{step} = \frac{|\text{Upper bound} - \text{Lower bound}|}{\text{Resolution}}. \tag{110}$$

**List of parameters.** Figs 1C, 9A and 5 and S3 Fig) resolution = 1000, correction-constant = 1, size = 1000, max(R) = 0.3, lower bound = -0.2, upper bound = 1.2. S1 Fig) resolution = 1000, correction-constant = 1, size = 1000, max(R) = 0.3, lower bound = -0.05, upper bound = 1.05. Fig 2A and 2B) resolution = 100, correction-constant = 1.1, size = 50, max(R) = 0.3, lower bound = -0.05, upper bound = 1.05. S2 Fig) resolution = 500, correction-constant = 1, size = 500, max(R) = 0.3, lower bound = -0.2, upper bound = 1.2. S5 Fig) resolution = 500, correction-constant = 1, size = 200, max(R) = 0, lower bound = -0.2, upper bound = 1.2.

**Two correlated patterns with adaptation and periodic inhibition.** In order to solve the dynamical equations of the mean-field in the presence of adaptation and global inhibition (as

done in S3 and S4 Figs) we compute at each point in time $\bar{\phi}_{x^1x^2}(m^1, m^1, \Theta_{x^1x^2})$, $p$, $q$, $R$ and $J_0(t)$. In particular, the four $\bar{\phi}_{x^1x^2}(m^1, m^1, \Theta_{x^1x^2})$ are solved first and recursively since they are functions of themselves. We then update $\Theta_{x^1,x^2}(t)$ with Euler method. Finally, we compute $m^1(t)$, $m^2(t)$. In order to compute $m^1(t)$, $m^2(t)$, we make a time-scale separation argument. We assume that $m^1(t)$ and $m^2(t)$ dynamics are much faster than $\Theta_{x^1,x^2}(t)$ and $J_0(t)$, $\tau \ll T_{J_0} < T_\theta$. According to this approximation, at each point in time we let $m^1(t)$ and $m^2(t)$ reach their equilibrium values given the current $\Theta_{x^1,x^2}(t)$ and $J_0(t)$. In other words, at each point in time, we consider all dynamical quantities frozen, than let $m^1(t)$ and $m^2(t)$ evolve according to their dynamics (we use Euler method) until convergence, and finally update the other quantities.

To find the fixed points in S3 Fig, we proceed like in the non-adaptive case: we do a grid search in the space spanned by $m^1$, $m^2$ and $R$. For each solution of $R$, $\bar{\phi}_{x^1x^2}(m^1, m^1, \Theta_{x^1x^2})$ are computed recursively. Finally, for the obtained values of $R$ and $\bar{\phi}_{x^1x^2}(m^1, m^1, \Theta_{x^1x^2})$, the solutions of $m^1$ and $m^2$ are found.

**Excluding self-interaction: A numerical approximation.** In Fig 2B, we compute the critical correlation for non-zero network load, $\alpha > 0$, in the case we consider the correction to exclude self interaction. To find the numerical solutions of the fixed points, we have approximated the input term $h(x^1, x^2, z)$ to the first order in $z$ as follows:

$$h(x^1, x^2, z) = \langle h(x^1, x^2, z)\rangle_z + A\sqrt{\alpha r}z. \tag{111}$$

Then the quantity

$$\langle h(x^1, x^2, z)\rangle_z = Ar_{\max}(x^1 - \gamma)m^1 + Ar_{\max}(x^2 - \gamma)m^2 + \left\langle \frac{A^2q\alpha\phi(h(x^1, x^2, z))}{(1 - Aq)}\right\rangle_z \tag{112}$$

can be approximated by

$$\langle h(x^1, x^2, z)\rangle_z \sim Ar_{\max}(x^1 - \gamma)m^1 + Ar_{\max}(x^2 - \gamma)m^2 + \frac{A^2q\alpha\phi(\langle h(x^1, x^2, z)\rangle_z)}{(1 - Aq)} \tag{113}$$

which is equivalent to take the order 0 term into the Taylor expansion of $h(x^1, x^2, z)$ for small $z$.

**Stability of the fixed points.** The stability of the fixed points in Figs 9, S5 and S2 is obtained by computing the Jacobian matrix of the differential equations for $m^1$ and $m^2$ from Eqs (43a) and (43b) respectively. Analogously, the stability of the fixed points in S2 and S3 Figs are obtained by computing the Jacobian matrix of the differential equations for $m^1$ and $m^2$ from Eq (68ca-b).

When the steepness of the transfer function is very high, $b > 1000$, we approximate the transfer function with an Heaviside. The system Eq (43) as well as the Jacobian matrix are rewritten in a simpler way for $b \to \infty$ as can be found in Eq (78).

In the numerical computation of the Jacobian matrix computed in Section "Stability of the fixed points", we exploited the symmetries under exchange of $m^1$ and $m^2$, for example $J_{22}(m^1, m^2) = J_{11}(m^2, m^1)$ and so on.

## Supporting information

**S1 Text. Supplementary text for "When shared concept cells support associations: Theory of overlapping memory engrams".**
(PDF)

**S1 Fig. Evolution of $m^1(t)$ and $m^2(t)$ according to the mean-field dynamics for supercritical correlation.** The system is initialized in the rest state. During the stimulation period (0.5–8s) $m^1(t)$ receives external input. A) The system state is plotted in the phase-plane before,

during and after stimulation respectively. B) The delay between the activation of $m^1(t)$ and $m^2(t)$ is highlighted. Parameters: $\gamma = 0.002$, $\hat{b} = 100$, $\hat{h}_0 = 0.25$, $r_{max} = 1$, $\hat{\tau} = \tau = 1 r_{max}$, $\alpha = 0$, $C = 0.2$.
(TIF)

**S2 Fig. Estimation of the correlation range in which retrieval of a chain of concepts is possible.** A) Estimation of the maximum correlation, which correspond to the loss of the two single retrieval states, when $J_0$ is lowest. B) Estimation of the minimum correlation, which corresponds to the creation of the stable fixed point at $m^1 = m^2 > 0$, when the inhibition $J_0$ is at its maximum. In both A and B adaptation is frozen and $\theta = 0$. Parameters: $\gamma = 0.002$, $\alpha = 0$, $\hat{b} = 50$, $\hat{h}_0 = 0$, $\min(\hat{J}_0) = 0.7$, $\min(\hat{J}_0) = 1.2$.
(TIF)

**S3 Fig.** A) Dynamical mean-field solutions for $m^1$ and $m^2$ in the case of two independent patterns. B) Phase planes corresponding to the minimum ($J_0 = 0.7$) and maximum ($J_0 = 1.2$) value of inhibition in the case of two independent patterns. C,D) Same as A and B, but for correlated patterns $C = 0.2$. Parameters in A—D: $\gamma = 0.1$, $\alpha = 0$, $b = 100$. E, F) Same as C and D but in the low activity regime and for independent patterns. G, H) Same as C and D but in the low activity regime. Parameters in E—H: $\gamma = 0.002$, $\alpha = 0$, $b = 100$, $\tau_\theta = 45$, $T = 0.015$, $T_{J_0} = 25$. For the dynamics: resolution = 200, factor = 1. For the phase-planes: resolution = 1000, factor = 1, upper bound = 1.2, lower bound = -0.2 (same as Figs 2 and 4).
(TIF)

**S4 Fig.** Retrieval dynamics in the presence of adaptation according to the mean-field equations (dashed lines), and comparison with Fig 4A (shaded solid lines) A) Only two patterns are correlated. B) Four patterns are correlated. Parameters: $N = 10^4$, $P = 16$ in full network simulations and $\alpha = 0$ in mean-field. $\gamma = 0.002$, $\tau_\theta = 45$, $T = 0.015$, $T_{J_0} = 25$ in both.
(TIF)

**S5 Fig. Equivalent of Fig 9 but with the parameters extracted from [24].** In A and B the transfer function parameters are taken as those of function $\phi$ in [24]: $A = 3.55$, $r_{max} = 76.2$, $b = 0.82$, $h_0 = 2.46$. On the other hand, in C and D I estimated the parameters of a Sigmoid function that fits the function $f(\phi)$ in [24] as follows: $A = 3.55$, $r_{max} = 0.83$, $b = 4.35$, $h_0 = 1.7$. In all plots $\gamma = 0.001$. A and C) The phase-plane for $c = 0$ shows the position of fixed points. B and D) Bifurcation diagram and critical fraction of shared neurons according to different parameter choices.
(TIF)

**S6 Fig. Comparison between model prediction and data.** Probability of finding a neuron responding to a given number of concepts as measured from experimental data (black stars), predicted by the three algorithms (as in Fig 6, the area between error bar of one standard deviation is shaded) and theoretically forecast for the indicator neuron model (light blue) and for the hierarchical generative model (light green) obtained from Eq (99). The theoretical predictions are not smooth curves due to choice of matching the subgroups sizes to the dataset.
(TIF)

## Acknowledgments

We thanks to Valentin Marc Schmutz, Martin Barry, Alireza Modirshanechi and Johanni Brea for useful comments and discussions.

## Author Contributions

**Conceptualization:** Chiara Gastaldi, Tilo Schwalger, Rodrigo Quian Quiroga, Wulfram Gerstner.

**Data curation:** Emanuela De Falco.

**Formal analysis:** Chiara Gastaldi, Tilo Schwalger, Emanuela De Falco, Wulfram Gerstner.

**Funding acquisition:** Wulfram Gerstner.

**Investigation:** Chiara Gastaldi.

**Methodology:** Chiara Gastaldi, Tilo Schwalger, Rodrigo Quian Quiroga, Wulfram Gerstner.

**Software:** Chiara Gastaldi.

**Supervision:** Tilo Schwalger, Wulfram Gerstner.

**Validation:** Tilo Schwalger, Emanuela De Falco, Rodrigo Quian Quiroga, Wulfram Gerstner.

**Writing – original draft:** Chiara Gastaldi, Tilo Schwalger, Rodrigo Quian Quiroga, Wulfram Gerstner.

**Writing – review & editing:** Chiara Gastaldi, Tilo Schwalger, Emanuela De Falco, Rodrigo Quian Quiroga, Wulfram Gerstner.

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
