## [Decision Letter · Decision Letter 0]

12 Jul 2021

Dear Mrs Gastaldi,

Thank you very much for submitting your manuscript "When shared concept cells support associations: theory of overlapping memory engrams" for consideration at PLOS Computational Biology. As with all papers reviewed by the journal, your manuscript was reviewed by members of the editorial board and by several independent reviewers. The reviewers appreciated the attention to an important topic. Based on the reviews, we are likely to accept this manuscript for publication, providing that you modify the manuscript according to the review recommendations.

Sincerely,

Abigail Morrison

Associate Editor

PLOS Computational Biology

Lyle Graham

Deputy Editor

PLOS Computational Biology

[LINK]

Reviewer's Responses to Questions

**Comments to the Authors:**

Reviewer #1: The authors use a computational model of attractor neural network for modeling association between concepts through shared concept cells. Their main findings are: (i) there is a minimum fraction c_min of shared cells, below which assemblies can't be recalled simultaneously, i.e. no association is created between the two concepts; (ii) there is a maximum fraction of shared cells c_max, above which assemblies can't be recalled individually, i.e. the two concepts are merged into one; (iii) c_min exists in the presence of global inhibition, and its value should be above chance level; (iv) the value of c_max depends on the threshold and steepness of the frequency-current curves, and does not depend on memory load; (v) in the presence of a periodically modulated background signal, the recall takes the form of association chains; (vi) predictions of a non hierarchical iterative overlap generating model match experimental data on the number of concepts to which a neuron responds, suggesting MTL encodes memory engrams in a non hierarchical way.

I find the results interesting and relevant for the PLOS CB audience. It is also well organized and clearly written.

Some minor comments:

Section - Association chains

It would be interesting to have a bit more information about the reactivation. Is there an order? What defines it? When does the cycle closes? Is it clear what defines the length/order of a cycle?

Section - How does a network embed groups of overlapping memories?

Motivation for the 3 different algorithms is not very clear. Hierarchical vs. non-hierarchical is clear, but why using two non-hierarchical, for example, is not well motivated. Brief description/intuition of the 3 different algorithms on results could help.

Section - Robustness to heterogeneity

What happens to the model predictions for further dilution of the weight matrix (d<0.8)? Some discussion about this would be interesting, considering the lower connection probability found in CA3 networks (Guzman et al., Science, 2016. DOI: 10.1126/science.aaf1836).

Typos:

Caption Figure 4C: "By decreasing their mean activity γ,"

Line 379: "transitions could also triggered"

Line 418: "can be interpret"

Reviewer #2: I reviewed a previous version of this manuscript (in thesis chapter form). The (minor) issues I brought up have been resolved in this version. I report below my previous summary of the findings and include suggestions for a couple of possible additional discussion points.

The focus of this work is related to the experimental observation of “concept” cells, discovered by recording neurons from the medial temporal lobe (MTL) of human subjects suffering from epilepsy. These neurons are active whenever specific abstract information - e.g., a specific person or object - is presented to or retrieved by a subject. In several experiments from various labs, it has been estimated that the representation of a given concept in MTL circuits is very sparse, only a small fraction of neurons is active on average for a given concept. These observations, and several previous other findings related to sparse memory representations of other stimuli in other species, have driven a considerable amount of research in artificial neural networks capable of storing and retrieving many sparse memory items. In these networks, memory items are represented by the activation of a small group of neurons. The choice of which neuron encodes which item is decided, independently for each neuron and each item, by tossing a biased coin. This bias determines the sparsity of the representations. The synaptic weight between every pair of neurons is then determined according to their desired activation across all the items. With an appropriate choice of the weights, the activation of a group of neuron becomes an attractor of the network dynamics.

The questions addressed in this work arise from the discrepancies between the experimental observations and the models. The standard assumption of random and uncorrelated sparse encoding of patterns in attractor models has an immediate consequence for the conjunctive encoding of multiple items. For instance, if a neuron participates to the encoding of a memory item with probability p, the probability that the same neuron will encode two memory items is p^2. This basic observation is at odds with empirical findings from statistical estimates of the encoding of concept cells. In the experiments, pairs of concepts have a higher probability of being encoded by a concept cell compared to the chance level defined above, especially if the two concepts are related (e.g., semantically). As a first question, the authors hence asked if and how attractor networks can deal with the higher than chance correlation between representations of distinct items. The problem is systematically attacked with a mathematical analysis of a new model that incorporates these extra correlations. First, the scenario in which two correlated patterns are stored in an attractor network is analyzed. The problem is studied with a mean-field description of the system, corroborated by numerical solutions of the full network dynamics. As a result, the maximal amount of correlation between two patterns that would still allow the network to retrieve the individual patterns, as opposed to only a mixture of the two, could be characterized. From this complete description of the possible dynamical regimes for two correlated patterns, the authors moved to examine several variations of the problem within the theory: (i) two correlated patterns stored together with many other uncorrelated patterns (ii) several stored patterns where distinct pairs are correlated (iii) correlations among more than two patterns. This is a thorough characterization of the possible states of networks storing correlated patterns in a variety of conditions, and it also shows that the estimated sparsity and correlations in concept cells are within the bound computed in the paper. This work shows that the standard attractor network models can be extended to account for interesting electrophysiological measurements in humans.

A second question addressed in the paper arises from another set of experimental and theoretical observations related to human memory. In free recall experiments, human participants are presented with a list of words, one word at a time. The participants are then asked to retrieve as many words as possible, in no particular order. There are a number of interesting systematic effects in the way the words are recalled in these experiments. Many of these effects can be explained by a variant of the standard attractor networks described above, where some additional dynamics (e.g., periodic modulation of inhibition or adaptation) induce transitions between the attractors in the networks, mimicking the sequential retrieval of words in free recall. In these models, the transitions are determined by the correlation between representations, which are assumed to arise from the random overlaps between the random uncorrelated representations. This mechanism relies on the finite size of the network models (and the not so sparse representations used in these models). The second question addressed in the paper is if and how the transition between the attractors would occur with very sparse, but correlated, patterns. In particular, the lower bounds on correlations that would allow the network dynamics to transition between attractors have been examined. The study of this problem allowed to show that the empirical estimates of correlation would also allow the network to perform retrieval in a way that could account for the results from free recall experiments.

Lastly, the paper examines three different models for generating correlated patterns (including models that rely on hierarchical representations of concepts). The resulting probability distributions for a neuron to be active for k different concepts are then compared with experimental data. The results argue against a hierarchical encoding in the MTL recordings. I particularly appreciate the effort of linking the theory with experimental data.

In summary, this work advances our understanding of computations with discrete attractors in recurrent neural networks. I think that this work is a novel useful contribution to the study of memory processes in theoretical neuroscience. This work also strengthens the link between theoretical and experimental approaches to the study of memory phenomena.

I recommend publication.

Below a couple of points that the authors might want to consider:

- The paper primarily examines correlations between disjoint pairs of stored items. It could be useful to see a discussion about scenarios where any pair has above chance correlations (e.g., some of the neurons tend to be active almost regardless of the identity of the memory item). Some concrete examples in hippocampus, in the context of spatial coding, can be found in (Rich, Liaw & Lee, 2014; Grosmark & Buzsaki, 2016) among others.

- The paper examines associative transitions between memory items in the presence of correlations, motivated by the experimental and modeling work on free recall. One prominent of feature of free recall is the sublinear scaling of the number of retrieved items vs the number of presented items. It would be useful to see a discussion about these scaling laws within the context of the theory proposed by the authors.

- In Discussion, the context-dependent disentanglement of memories could be compared and contrasted with the work described in (Podlaski, Agnes & Vogels, bioRxiv 2020).

Minor: “backgrong” typo in Fig 2B legend

**Have the authors made all data and (if applicable) computational code underlying the findings in their manuscript fully available?**

Reviewer #1: Yes

Reviewer #2: **No: **I have not seen a link to data and code, apologies if it's there and I missed it.

PLOS authors have the option to publish the peer review history of their article (what does this mean?). If published, this will include your full peer review and any attached files.

Reviewer #1: No

Reviewer #2: No

Figure Files:

Data Requirements:

Reproducibility:

References:

---

## [Editor Report · Decision Letter 1]

29 Nov 2021

Dear Mrs Gastaldi,

We are pleased to inform you that your manuscript 'When shared concept cells support associations: theory of overlapping memory engrams' has been provisionally accepted for publication in PLOS Computational Biology.

Best regards,

Abigail Morrison

Associate Editor

PLOS Computational Biology

Lyle Graham

Deputy Editor

PLOS Computational Biology

---

## [Editor Report · Acceptance letter]

16 Dec 2021

PCOMPBIOL-D-21-00730R1 

When shared concept cells support associations: theory of overlapping memory engrams

Dear Dr Gastaldi,

I am pleased to inform you that your manuscript has been formally accepted for publication in PLOS Computational Biology. Your manuscript is now with our production department and you will be notified of the publication date in due course.

With kind regards,

Olena Szabo
